# PathIntegrate: Multivariate modelling approaches for pathway-based multi-omics data integration

**Cecilia Wieder**[1], **Juliette Cooke**[2], **Clement Frainay**[2], **Nathalie Poupin**[2],
**Russell Bowler**[3], **Fabien Jourdan**[4], **Katerina J. Kechris**[5], **Rachel PJ Lai**[6],
**Timothy Ebbels**[1] *

**1** Section of Bioinformatics, Division of Systems Medicine, Department of Metabolism, Digestion, and Reproduction, Faculty of Medicine, Imperial College London, London, United Kingdom, **2** Toxalim (Research Centre in Food Toxicology), Université de Toulouse, INRAE, ENVT, INP-Purpan, UPS, Toulouse, France, **3** National Jewish Health, Denver, Colorado, United States of America, **4** MetaboHUB-Metatoul, National Infrastructure of Metabolomics and Fluxomics, Toulouse, France, **5** Department of Biostatistics and Informatics, Colorado School of Public Health, University of Colorado Anschutz Medical Campus, Aurora, Colorado, United States of America, **6** Department of Infectious Disease, Faculty of Medicine, Imperial College London, London, United Kingdom

* t.ebbels@imperial.ac.uk

**Data Availability Statement:** The COVID dataset is publicly available from Mendeley data (https://data.mendeley.com/datasets/tzydswhhb5/5). The COPDgene multi-omics data can be found at the following sources: Clinical Data and SOMAScan

## Abstract

As terabytes of multi-omics data are being generated, there is an ever-increasing need for methods facilitating the integration and interpretation of such data. Current multi-omics integration methods typically output lists, clusters, or subnetworks of molecules related to an outcome. Even with expert domain knowledge, discerning the biological processes involved is a time-consuming activity. Here we propose PathIntegrate, a method for integrating multi-omics datasets based on pathways, designed to exploit knowledge of biological systems and thus provide interpretable models for such studies. PathIntegrate employs single-sample pathway analysis to transform multi-omics datasets from the molecular to the pathway-level, and applies a predictive single-view or multi-view model to integrate the data. Model outputs include multi-omics pathways ranked by their contribution to the outcome prediction, the contribution of each omics layer, and the importance of each molecule in a pathway. Using semi-synthetic data we demonstrate the benefit of grouping molecules into pathways to detect signals in low signal-to-noise scenarios, as well as the ability of PathIntegrate to precisely identify important pathways at low effect sizes. Finally, using COPD and COVID-19 data we showcase how PathIntegrate enables convenient integration and interpretation of complex high-dimensional multi-omics datasets. PathIntegrate is available as an open-source Python package.

## Author summary

Omics data, which provides a readout of the levels of molecules such as genes, proteins, and metabolites in a sample, is frequently generated to study biological processes and

data are available through COPDGene (https://www.ncbi.nlm.nih.gov/gap/, ID: phs000179.v6.p2). RNA-Seq data is available through dbGaP (https://www.ncbi.nlm.nih.gov/gap/, ID: phs000765.v3.p2). Metabolon data is available at Metabolomics Workbench (https://www.metabolomicsworkbench.org/ ID: PR000907). PathIntegrate is available via the open-source PathIntegrate Python package (www.github.com/cwieder/PathIntegrate). Tutorials and documentation for PathIntegrate can be found at https://cwieder.github.io/PathIntegrate. Source code for benchmarking and applications can be found at https://github.com/cwieder/PathIntegrate_scripts.

**Funding:** CW, TE - This research was funded in whole, or in part, by the Wellcome Trust [222837/Z/21/Z]. For the purpose of open access, the author has applied a CC BY public copyright licence to any Author Accepted Manuscript version arising from this submission. TE acknowledges partial support from BBSRC grants BB/T007974/1 and BB/W002345/1. RPJL was supported by a UK MRC fellowship (MR/R008922/1) which is part of the EDCTP2 programme supported by the European Union and a NIH-NIAID grant (R01 AI145436). JC is supported by a state-funded PhD contract (MESRI (Minister of Higher Education, Research and Innovation)). FJ - This research was funded by the Agence Nationale de la Recherche (ANR, French National Research Agency)—MetaboHUB, the national metabolomics and fluxomics infrastructure (Grant ANR-INBS-0010). KK, RB - Research reported in this publication was supported by the National Heart Lung Blood Institute of the National Institutes of Health to KK and RB under award number R01HL152735. This work was supported by NHLBI grants U01 HL089897 and U01 HL089856 and by NIH contract 75N92023D00011. The COPDGene study (NCT00608764) is also supported by the COPD Foundation through contributions made to an Industry Advisory Committee that has included AstraZeneca, Bayer Pharmaceuticals, Boehringer-Ingelheim, Genentech, GlaxoSmithKline, Novartis, Pfizer, and Sunovion. The content is solely the responsibility of the authors and does not necessarily represent the official views of the National Heart, Lung, and Blood Institute or the National Institutes of Health. The funders did not play any role in the study design, data collection, analysis, or publication of this work.

**Competing interests:** The authors have declared that no competing interests exist.

perturbations within an organism. Combining multiple omics data types can provide a more comprehensive understanding of the underlying biology, making it possible to piece together how different molecules interact. There exist many software packages designed to integrate multi-omics data, but interpreting the resulting outputs remains a challenge. Placing molecules into the context of biological pathways enables us to better understand their collective functions and understand how they may contribute to the condition under study. We have developed PathIntegrate, a pathway-based multi-omics integration tool which helps integrate and interpret multi-omics data in a single step using machine learning. By integrating data at the pathway rather than the molecular level, the relationships between molecules in pathways can be strengthened and more readily identified. PathIntegrate is demonstrated on Chronic Obstructive Pulmonary Disease and COVID-19 metabolomics, proteomics, and transcriptomics datasets, showcasing its ability to efficiently extract perturbed multi-omics pathways from large-scale datasets.

## Introduction

Multi-omics data integration is rapidly becoming a mainstream strategy used to elucidate complex molecular mechanisms in biological systems. Data profiled using diverse modalities, including genomics, epigenomics, transcriptomics, proteomics, and metabolomics provides complementary insights into the regulation of diverse biomolecules and their cellular functions [1]. Multi-omics data integration can delineate the transition from genotype to phenotype, while providing a more holistic view of a biological system. Despite the promise that multi-omics integration holds, the field itself is relatively young and faces numerous challenges [1–6]. Among these is the question of which method to use, and how to interpret the results. Several review papers categorise multi-omics integration methods according to underlying concepts, models, or intended purposes [7]. The choice of method used will depend highly on the desired outcome, which can be broadly split into outcome prediction (e.g. sample stratification) or elucidating molecular mechanisms (but often a combination of these). Studies focused on outcome prediction may leverage integration methods based on kernels or deep learning to optimise predictive performance [8–10], whereas those where the goal is hypothesis generation may opt for more explainable models using classical supervised [11,12] or unsupervised learning approaches [12–15], joint pathway analysis [16–19], network models [12,20], or Bayesian statistics [7]. The latter 'hypothesis generation'-based analysis, regardless of the method used, will often output results in the form of lists of molecules (i.e. genes, proteins, metabolites), typically ranked by their contribution to the model. Depending on the parameters and outputs of the model, the end-user may have multiple latent variables [13], clusters [21,22], or networks [23] composed of many molecules (genes, proteins, and metabolites) to analyse. Doing so is not only time consuming but requires expert domain knowledge to place biomolecules into a functional context.

Pathway analysis (PA) refers to computational methods that have been specifically developed to alleviate the task of analysing long lists of molecules by placing them into a functional context based on curated pathway collections [24]. Generally, conventional PA methods such as over-representation analysis or gene set enrichment analysis use statistical tests to determine which pathways are associated with a phenotype of interest [25,26]. The output is typically a list of significantly enriched pathways and their associated test statistics and $p$-values. PA methods are frequently used due to their convenient representation of omics data in the form of pathway descriptors, providing a straightforward interpretation of the biological processes

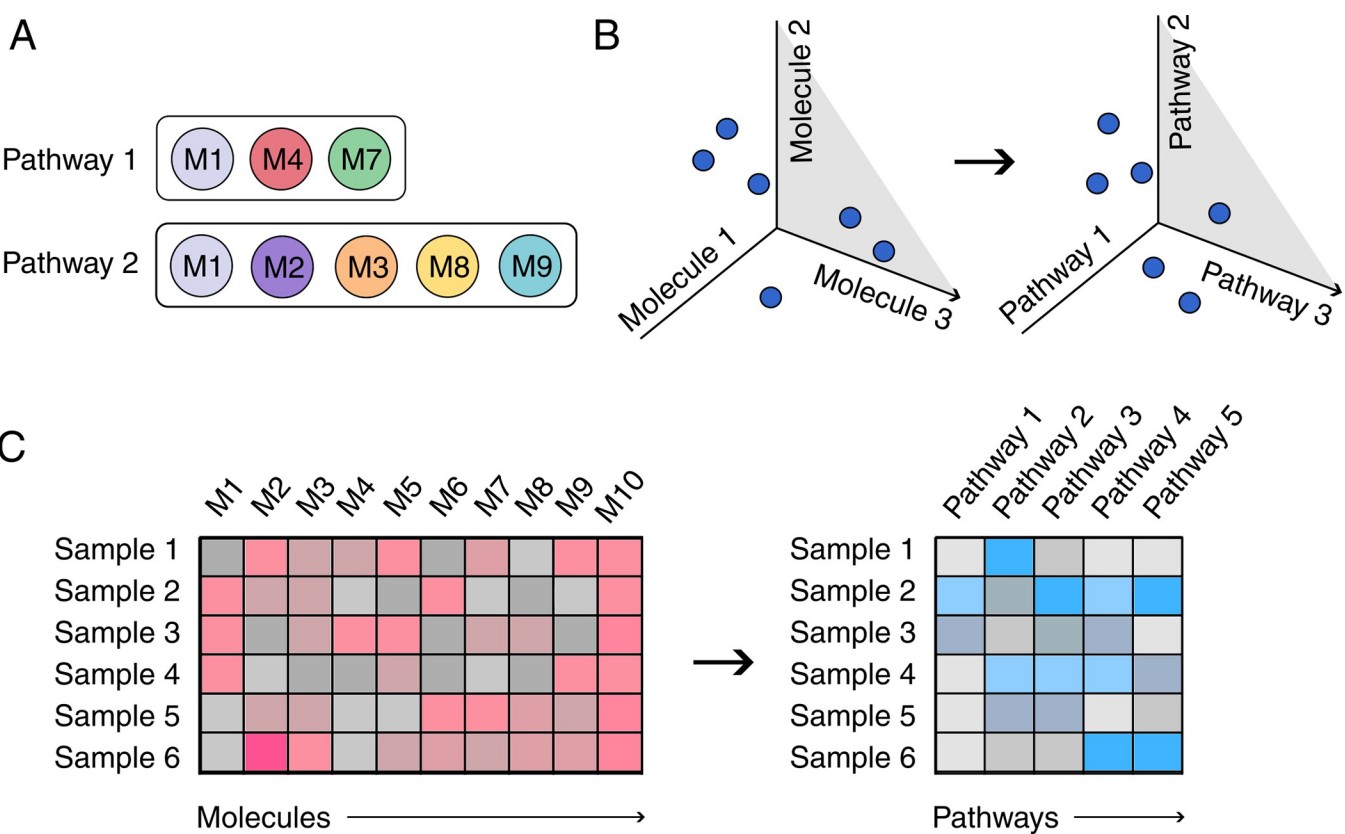

**Fig 1. Overview of pathway transformation using single sample pathway analysis (ssPA).** A. Pathways are represented as sets of molecules, e.g. genes, proteins, and metabolites. B) Pathway transformation by ssPA facilitates a change of dimension of an omics dataset from a molecular space to a pathway space. C) This transforms a sample-by-molecule expression or abundance matrix to a sample-by-pathway matrix, where values represent the 'activity' of each pathway for each individual sample.

that may contribute to disease phenotypes. Multi-omics pathway analysis is a relatively new but promising area of research [27]. Tools such as MultiGSEA [19], ActivePathways [17], PaintOmics [28], and IMPaLA [16] all leverage multiple layers of biological information to compute enrichment of multi-omics pathways, associated statistical significance levels, and visualisations as an end-result. While highly useful, these methods lack certain desirable features, including the ability to predict outcomes, enabling model performance evaluation, or obtaining a representation of the data in a lower-dimensional space. These goals can be achieved by using pathway-based predictive models, which use pathway rather than molecular-level features to model and predict new data, and infer pathway enrichment through feature importance [29–32]. We provide a detailed overview of related methods in supplementary information (Related work in S1 Supporting Information), but to the best of our knowledge, we are unaware of any one method which provides predictive, integrative modelling of multi-omics data at the pathway-level.

In this work we introduce PathIntegrate, a modelling framework and corresponding Python toolkit to facilitate pathway-based multi-omics integration. PathIntegrate employs single-sample pathway analysis approaches (ssPA) (Fig 1), which transform molecular-level abundance data matrices into pathway-level matrices, by using summarisation approaches (e.g. principal component analysis (PCA)) to condense molecular-level measurements into pathway scores for each individual sample in a dataset [33–37]. By using pathway-transformed

multi-omics datasets as input to multivariate supervised models, multi-omics data can be integrated at the pathway-level, providing the user with a range of outputs including i) interpretation of multi-omics pathways associated with the outcome, ii) prediction of outcomes, iii) contribution of each omics view to the model and prediction (in the case of multi-view models), iv) projection of the multi-omics data to a lower dimensional space (in the case of latent variable models). An inherent challenge in multi-omics integration is the heterogeneity between omics datatypes, both in terms of the number of features profiled and the range of numerical values. PathIntegrate substantially contributes to addressing these with the pathway-transformation step, where disparate omics datasets are brought to a common scale, i.e. in terms of pathway 'activity'. Compared to their molecular-level counterparts, pathway-based multi-omics integration models can provide a more parsimonious model when there are fewer input pathways than molecules, while also enabling the detection of multiple small, correlated signals that may not be detected in the molecular-level data. Moreover, pathway-based modelling could increase robustness to data noise by maximising biological variation and simultaneously reducing technical variation [29].

PathIntegrate consists of two supervised learning frameworks for pathway-based multi-omics integration: PathIntegrate Single-View, which produces a multi-omics pathway-transformed dataset and applies a classification or regression model to the data, and PathIntegrate Multi-View, which uses a multi-block partial least regression (MB-PLS) model to model interactions between pathway-transformed omics datasets. Note that both PathIntegrate Multi-View and Single-View are multi-omics integration methods, and here we use the terms 'Multi' and 'Single' to refer to the type of predictive model applied (multi-view or single-view [38]). As both these frameworks rely on pathway transformation (ssPA) of the input omics data, we first demonstrate the ability of univariate methods to detect pathway signals at higher power than molecular-level signals in low signal-to-noise scenarios. We then show that PathIntegrate models can precisely detect enriched pathways even at low effect sizes, as well as use this information to accurately classify samples. PathIntegrate was benchmarked against DIABLO [11], a popular multi-omics integration tool with a similar predictive framework, but which does not use pathway transformation. Finally, we showcase the benefits of using PathIntegrate to interpret complex data using case studies on Chronic Obstructive Pulmonary Disease (COPD) and COVID-19 multi-omics datasets, illustrating the ability of the method to identify important and relevant pathway signatures. The PathIntegrate Python package is freely available at https://github.com/cwieder/PathIntegrate, and is designed to be compatible with many SciKit-Learn [39] functions, enabling fast and efficient model optimisation and evaluation. PathIntegrate models are fitted in minutes and can run on a laptop with standard hardware (e.g. 8GB RAM, 1.4 GHz processor).

## Results

### Pathway transformation increases sensitivity to coordinated, low signal-to-noise biological signals

Aside from improvements in interpretability, we hypothesized that pathway-based modelling or transformation of data can also provide increased sensitivity in detection of pathway signals in the data, particularly in low signal-to-noise scenarios. By combining abundance levels of correlated individual molecules within a pathway, we anticipate that statistical methods will be able to detect the pathway signal with higher power than individual molecular signals alone. Throughout this work, we refer to 'molecular-level' models as those with individual molecular entities (such as genes, proteins, and metabolites) as input features, as opposed to 'pathway-level' models, which take ssPA pathway-transformed data as input and hence features

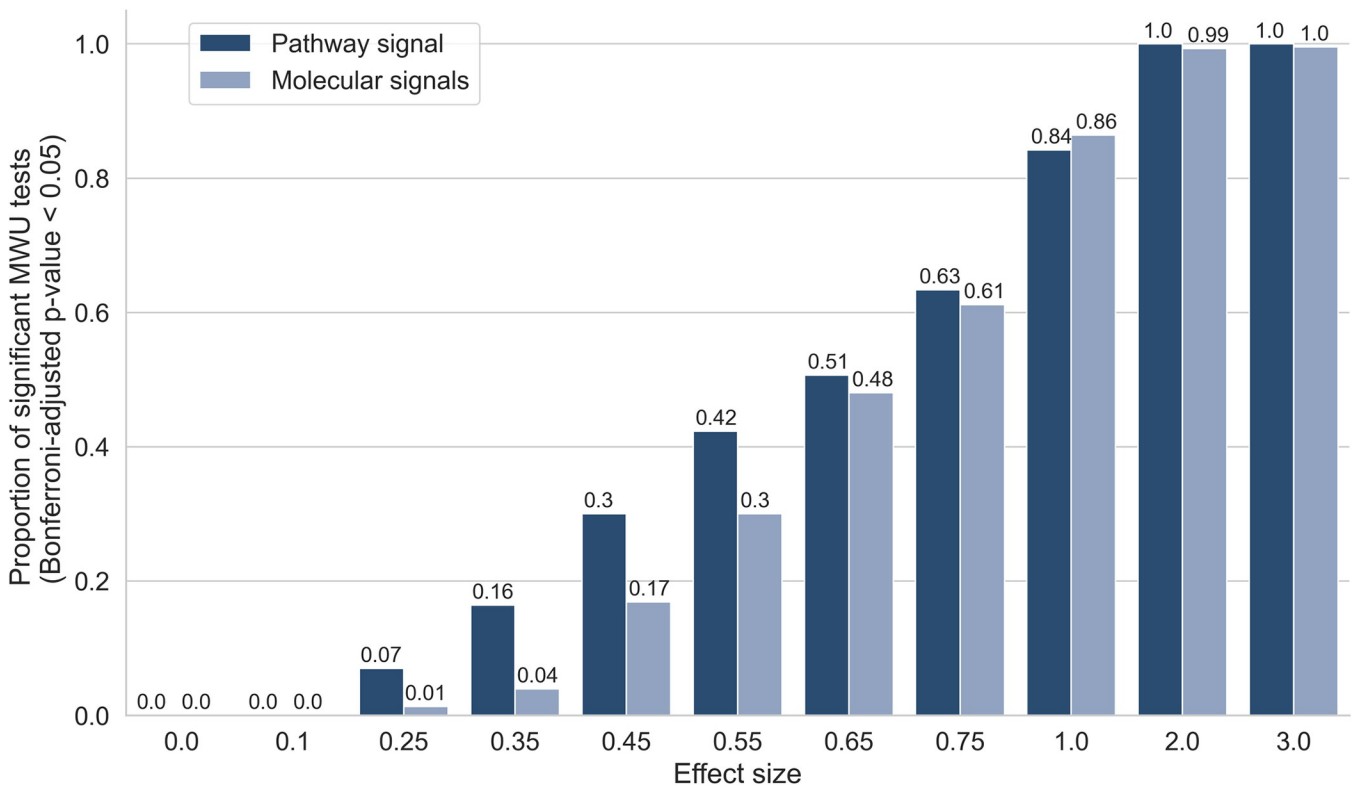

**Fig 2. Pathway transformation enhances sensitivity to low signal-to-noise signals.** y axis shows proportion of MWU tests significant at Bonferroni $p \leq 0.05$, performed either on the pathway-level data or the molecular level data, at varying effect sizes shown on x-axis. Semi-synthetic data based on COVID-19 dataset.

represent a combination of molecules in each pathway. Briefly, ssPA methods require an $X_{N \times M}$ matrix of molecules as input and combine the abundance values of molecules in a set of predefined pathways to provide an $A_{N \times P}$ pathway-level matrix, where features represent pathways and each sample has an 'activity score' for each pathway (see Methods).

The use of 'semi-synthetic' data, in which artificial biological signals are inserted into experimental multi-omics data, provides us with a ground truth we can use to benchmark methods throughout this work [33]. We used semi-synthetic multi-omics (metabolomics and proteomics) data derived from COPD and COVID-19 studies (see Methods) to examine whether pathway transformation of multi-omics data allowed pathway signals to be detected by univariate analysis (Mann Whitney-U tests (MWU)) at higher power than individual molecular signals (Fig 2 and Fig C in S1 Supporting Information). Each omics dataset was transformed to the pathway level using ssPA, using the kPCA ssPA method [33] (see Methods). At each realisation of the simulation, repeated for each Reactome pathway accessible in the datasets, we enriched all the molecules in the pathway (metabolites and/or proteins) in the simulated disease group for a range of effect sizes, corresponding to the range of $\log_2$ fold changes observed in the original datasets (Fig A and Fig B in S1 Supporting Information).

We applied MWU tests to detect differences between the simulated phenotype groups based on the enrichment of each of the individual molecules in the molecular level data or ssPA scores of the target pathway itself. For the molecular level simulation, we applied Fisher's method to combine p-values in the target pathway if at least 50% constituent molecules were significant ($p \leq 0.05$), otherwise the combined $p$-value was set to 1. Encouragingly, at lower effect sizes (i.e. 0.25–0.55), we observed a higher proportion of significant $p$-values in the

pathway-transformed data than in the molecular level data. The same trends were observed irrespective of the dataset used to create the simulation (Fig 2 and Fig C in S1 Supporting Information). This suggests that pathway-transformation approaches could improve the detection of low signal-to-noise, correlated signals in multi-omics datasets, and motivates the use of PathIntegrate models in the remainder of this work, which use ssPA pathway transformation to enable pathway-based multi-omics integration.

## PathIntegrate: Supervised pathway-based multi-omics integration frameworks

In this study we present and investigate the use of the PathIntegrate modelling frameworks for multi-omics pathway-based integration (Fig 3). PathIntegrate provides two supervised models:

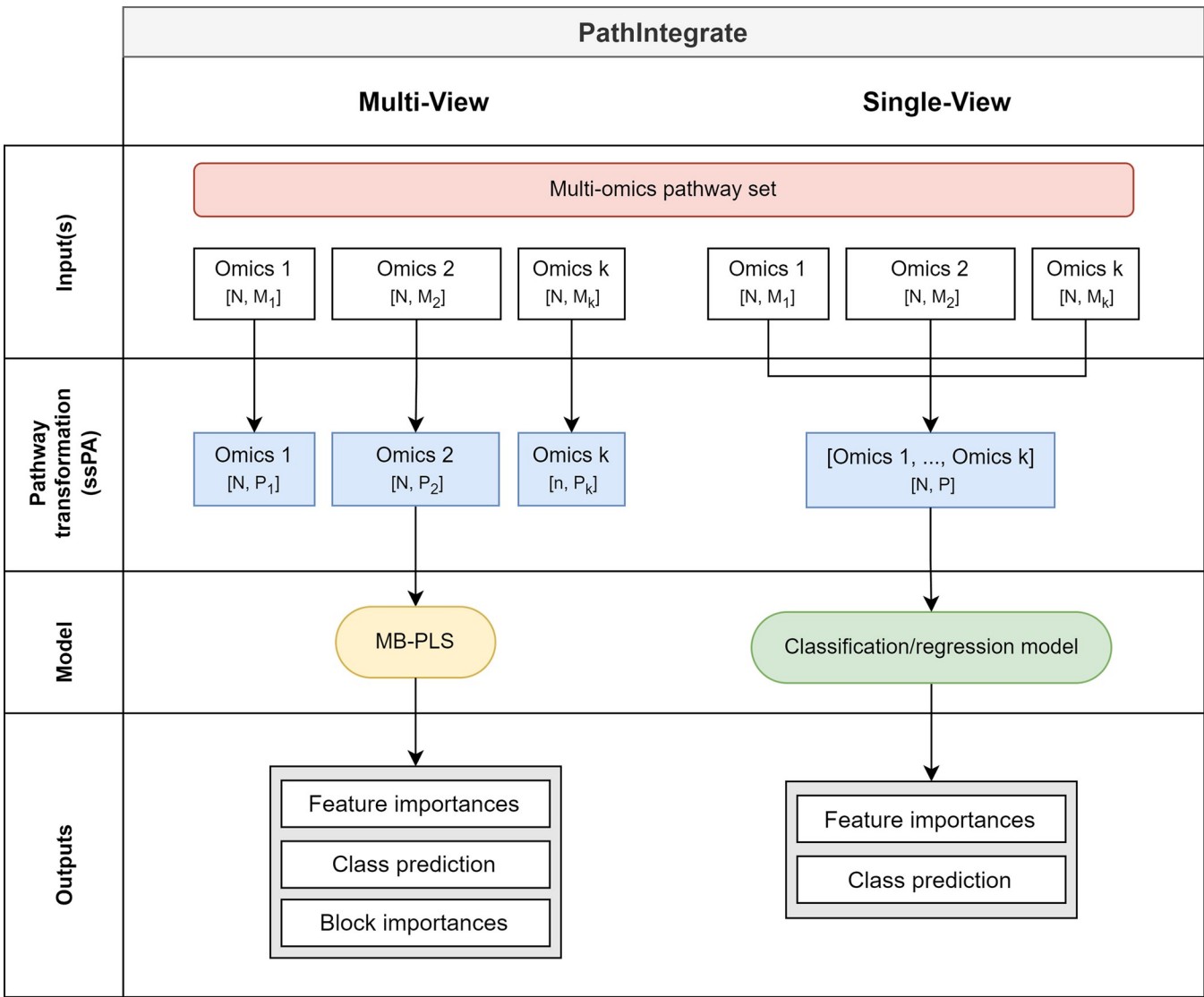

**Fig 3. PathIntegrate Multi-View (left) and Single-View (right) modelling frameworks for multi-omics pathway-based integration.** Frameworks are outlined in terms of their input data, pathway-transformation stage, statistical model, and outputs. Blue data blocks represent omics data which has been transformed from the molecular ($X_{N \times M}$) space to the pathway ($A_{N \times P}$) space using ssPA. Both Single-View and Multi-View make use of the same multi-omics pathway set.

Multi-View and Single-View. They are both designed to take two or more ($k$) $X_{N \times M}$ sample-by-molecule omics abundance matrices as well as a labelled outcome vector $y$ as input and apply a single-sample pathway analysis transformation (facilitated by our recently published ssPA Python package [33]) before a predictive model is applied to the data. PathIntegrate can model both continuous and binary outcomes using classification and regression models, but for simplicity we have demonstrated it using binary (e.g. case-control) outcomes throughout this work. Both frameworks achieve the same key outcomes: i) using pathway scores to predict an outcome, and ii) ranking multi-omics pathways by importance in the prediction. PathIntegrate Multi-View uses a multi-table integration model and can therefore provide interpretable insights both within and between omics views, whereas PathIntegrate Single-View provides more flexibility on the high-level predictive model applied and can be better tuned towards prediction. Both models use a single set of multi-omics pathways $P$, where each pathway has a unique identifier and description, and contains a set of molecular identifiers which can either belong to different omics (i.e. metabolites, proteins, and genes) or in some cases only one omics (i.e. only proteins). Using these pathways, PathIntegrate Multi-View computes pathway scores on each omics view separately, whereas Single-View computes them from multi-omics data.

PathIntegrate Multi-View uses a multi-block partial least squares (MB-PLS) latent variable model to integrate ssPA-transformed multi-omics data. Each omics block is transformed to the pathway level individually and the resulting $k$ $A_{N \times P_i}$ blocks are used as input to the MB-PLS model. This preserves the block structure of each omics view and importantly allows users to compute how much each view contributes to the prediction of the outcome variable $y$, as well as extract within- and between-omics level results such as pathway importances and latent variable representations (scores and superscores [40–42]). Importantly, the latent variable model used by Multi-View enables extraction of orthogonal biological effects, similar to PCA, possibly capturing contrasting processes. Furthermore, such models are ideal for pathway-level data, where there is expected to be a high degree of overlap and co-linearity which is accounted for by the PLS framework.

PathIntegrate Single-View begins by computing multi-omics pathway scores by performing ssPA transformation on molecular abundance or expression profiles obtained across multiple omics data blocks (e.g. genes, proteins, and metabolites). A single $A_{N \times P}$ pathway-level matrix is returned, in which each feature represents the 'activity' of each sample in a multi-omics pathway. The resulting multi-omics pathway scores are used as input to a predictive model (any SciKitLearn compatible model e.g., partial least squares discriminant analysis (PLS-DA), logistic regression, support vector machine, random forest, etc). Pathway importances can be obtained using variable selection approaches appropriate for the model used (e.g., Gini impurity for random forests or the $\beta$ coefficient for regression-based models).

By describing and evaluating the two PathIntegrate modelling frameworks we aim to help users select the method best suited to their study design and research questions.

## PathIntegrate performance evaluation

PathIntegrate Multi-View and Single-View were evaluated in a classification setting by a) the ability to discriminate between sample classes based on important pathways, and b) the ability to rank important pathways highly. Using semi-synthetic simulated metabolomics and proteomics data (see Methods) we enriched one target Reactome pathway containing metabolites and/or proteins at a time, at varying effect sizes, and repeated this for each pathway accessible in the datasets. For simplicity and consistency between datasets we integrated two omics throughout the performance evaluation section. Results based on COPDgene semi-synthetic

data are shown in Fig 4, and results based on COVID-19 semi-synthetic data are shown in Fig H in S1 Supporting Information. Note that this simulation design is rather conservative, because only one pathway is enriched in each realisation (although its constituent molecules may overlap with other pathways), whereas in a real biological system we may expect multiple pathways to be enriched at once. PathIntegrate Multi-View used multi-block PLS as the underlying predictive model, and for purposes of comparison, PathIntegrate Single-View used standard PLS-DA.

We compared PathIntegrate to the state-of-the-art multi-omics integration method DIABLO from MixOmics [11,43]. To the best of our knowledge, DIABLO is the most similar multi-omics integration method developed to date which makes use of a multi-view framework. As DIABLO is flexible as to the input data matrices, we compared standard DIABLO (using molecular-level omics data, 'DIABLO molecular-level'), as well as a pathway-based DIABLO ('DIABLO pathway') using the same ssPA-transformed omics matrices as input to PathIntegrate Multi-View. Importantly, although we are comparing the performance of PathIntegrate to DIABLO, we do not expect significant increases in predictivity or ability to detect the target pathway, due to the similarity of the underlying generalised canonical correlation analysis model to MB-PLS. Instead, we aim to highlight the flexibility of using pathway scores as input to supervised integrative models, such as DIABLO, and that even using different multivariate algorithms can yield predictive models capable of identifying target pathways with high sensitivity and specificity, and thus generating more interpretable results.

A fundamental question is whether modelling data using pathways can yield improvements in predictive performance compared to using molecular level data. Fig 4A shows the ability of PathIntegrate Multi-View, PathIntegrate Single-View, and DIABLO to predict samples in an unseen test set based on AUROC (Fig 4A, Fig H in S1 Supporting Information). All methods began to discriminate sample classes even at low effect sizes (0.1–0.25), concordant with findings from the univariate simulation. The pathway-based models (PathIntegrate Multi-View, Single-View and 'DIABLO pathway') exhibited improved performance compared to the 'DIABLO molecular-level' (standard) model across all effect sizes. As effect size increased from 0.25–1.0 the PathIntegrate methods performed similarly to 'DIABLO pathway'. Overall, these results suggest that using pathway-level models may yield improved predictive performance compared to molecular-level models.

We also compared the predictive performance of PathIntegrate models using pathways from two different databases, Reactome and KEGG, as well as the performance of MB-PLS and PLS models using the molecular-level data (i.e. PathIntegrate without the pathway-transformation step) (Fig 4C/Fig E in S1 Supporting Information shows PathIntegrate Multi-View and Fig H/Fig I in S1 Supporting Information show PathIntegrate Single-View and DIABLO). Results for the molecular level simulation can vary depending on the number of molecules enriched at each realisation, which correspond to the size of the pathway in the equivalent pathway-level simulation. Because Reactome and KEGG have differing distributions of pathway sizes [44], we randomly sampled the number of molecules enriched in each realisation based on the combined distribution of Reactome and KEGG pathway sizes, in order to reduce dependence on database pathway size. At lower effect sizes (0.1–0.25), both the molecular and pathway-level models performed similarly, whereas at moderate-to-high effects the pathway-based models exhibited an increase in predictive performance concordant with trends observed in Fig 4A. Models based on KEGG pathways appear to perform marginally better than Reactome pathways at larger effect sizes, which may be due to KEGG pathways being larger on average (see Fig D and Table A in S1 Supporting Information for pathway database size statistics).

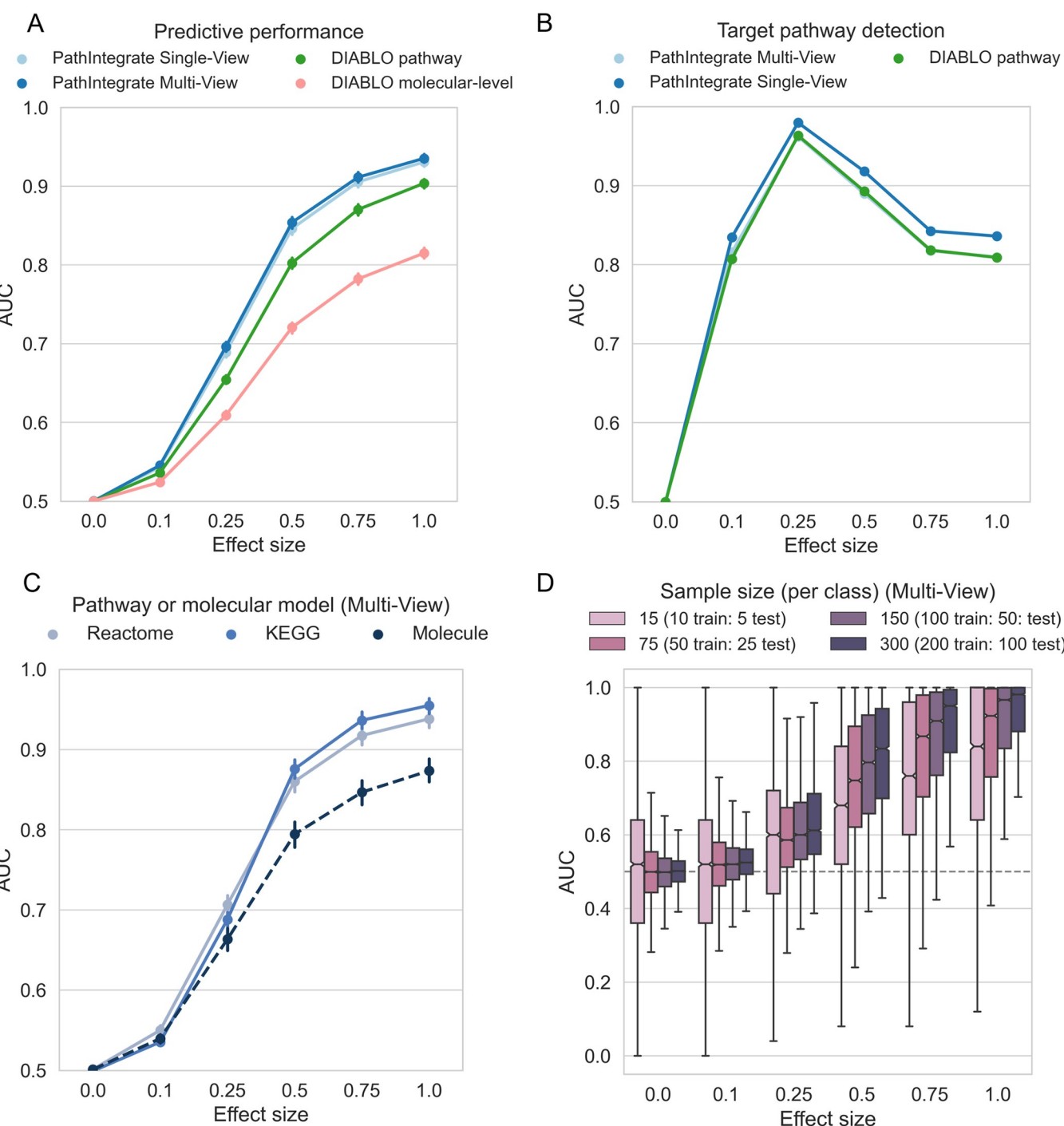

**Fig 4. Performance of PathIntegrate and DIABLO vs. effect size, based on semi-synthetic data measured by AUROC.** COPDgene metabolomics and proteomics data were integrated in each model. A. Ability to correctly predict sample outcomes (case vs. control). We compared PathIntegrate Multi-View and Single-View to DIABLO using both molecular and pathway-level multi-omics data. B. Ability to correctly recall target enriched pathway. We compared DIABLO RGCCA model loadings to the Multi-View MB-PLS VIP and Single-View PLS VIP statistics for pathway importance. C. Comparison of PathIntegrate Multi-View classification performance using KEGG and Reactome pathway databases as well as molecular-level model. D. Effect of sample size on PathIntegrate Multi-View classification performance. For panels a-c error bars indicate 95% confidence intervals on the mean AUROC (in some cases they appear smaller than point sizes).

We next evaluated the ability of PathIntegrate and 'DIABLO pathway' to accurately detect the target enriched pathway. (Fig 4B, Fig H in S1 Supporting Information). For PathIntegrate Single-View and Multi-View methods, variable importance in projection (VIP and multi-block-VIP) were used to evaluate feature importances[41]. $p$-values for the significance of each pathway feature VIP or MB-VIP value were computed empirically based on 10,000 sample permutations with BH-FDR correction. For 'DIABLO pathway', the RGCCA loadings on component 1 were used to infer feature importance, and $p$-values were subsequently computed using the same permutation testing approach. A true positive enriched pathway was defined as being the target enriched pathway and having an adjusted $p$-value of $\leq 0.05$ (see Methods for full description of the confusion matrix computation). Both PathIntegrate and DIABLO models performed well in terms of target pathway detection, even being able to detect the target pathway with high AUC ($\geq 0.90$) at low effect and high noise scenarios (effect size = 0.25). PathIntegrate Multi-View performed almost identically to 'DIABLO pathway'. All methods experience a decrease in AUC at higher effect sizes (0.5–1), which is expected due to pathways overlapping with the target pathway reaching significance, and in-built normalisation of the model weights/loadings causing the magnitude of the coefficient of the target pathway to shrink slightly in comparison to those of highly overlapping pathways. For simplicity, these overlapping pathways are treated as false positives, though they contain truly differentially abundant molecules. Thus, this decrease does not point to a lower performance of the method in identifying pathways relevant to prediction of the outcome. Furthermore, while the primary emphasis of this work is not on contrasting regularized and non-regularized models, it is worth noting that sparse models are widely used for feature selection. We also compared the ability of the models to select the target pathway with a sparse version of DIABLO (using the L1 norm, see Methods) (Fig F/Fig H in S1 Supporting Information). At low to moderate effect sizes, the sparse model identified the target pathway at similar AUC to the PathIntegrate/non-regularised DIABLO model, but at high effect sizes it showed slight improvements in target pathway identification as the sparsity constraint prevented high numbers of overlapping pathways reaching significance.

Finally, we investigated the effect of sample size, which is well known to influence model performance, on PathIntegrate models. We down-sampled each of the two classes in the data, keeping a 1:1 ratio between classes, and evaluated the predictive ability of the models at varying effect sizes (Fig 4D/Fig G in S1 Supporting Information and Fig H/Fig J in S1 Supporting Information show results for Multi-View and Single-View respectively). As expected, the lower the number of samples in the model, the more variability observed in the predictions. Particularly at lower effect sizes, smaller sample numbers were more likely to result in false positives and spurious results. While it is not possible to state the minimum number of samples necessary to apply PathIntegrate models, it is important for users to test the performance of the model using appropriate cross-validation approaches to be confident that the conclusions are statistically robust.

While these results demonstrate the predictive ability of PathIntegrate models, it is challenging to create a realistic simulation scenario which accurately reflects molecular activities and their participation in pathways in a biological system. Hence, we have applied PathIntegrate to the COPDgene and COVID-19 experimental datasets in the Application section to further illustrate model performance and interpretation.

## PathIntegrate Multi-View applied to COPDgene data

The COPDgene cohort consists of 10,198 smokers at baseline with and without chronic obstructive pulmonary disease (COPD) [45]. We integrated metabolomics, proteomics, and

transcriptomics multi-omics data measured at Phase 2 (~5 years after baseline) profiled on a subset of individuals with all three omics data (n = 522) using PathIntegrate to identify Reactome pathways associated with COPD pathology. The Multi-View model of PathIntegrate allows users to gain rich insights into the underlying data, from high-level interpretation of the global rankings of enriched pathways, to being able to investigate the importance of pathways in each omics block and latent component individually. We applied the kPCA ssPA method to produce pathway score matrices for each omics view and using 5-fold cross validation, we found that four latent variables yielded an optimised MB-PLS model (mean cross-validated AUC: 0.70) (Fig K in S1 Supporting Information). The MBPLS superscores for each of the four latent variables coloured by COPD status are shown in Fig 5A, providing a visual representation of the ability of multi-omics pathways to identify differences between COPD and non-COPD groups, in which each of the four latent variables exhibit a visible difference between groups.

One of the primary insights obtained from the Multi-View model is the contribution of each omics view to the variance explained in the outcome variable $y$ (Fig 5B). In the first latent variable, all three omics accounted for a considerable proportion of the variance explained in y, suggesting the pathway scores correlate well in the latent space. In the further three latent variables, transcriptomics and proteomics views tend to contribute most to the outcome prediction. Although metabolomics describes less of the variance in $y$ than the other omics, based on 100 bootstrap samples the mean variance explained across all latent variables remained between 6 and 17 percent. The dominance of transcriptomics and proteomics views may suggest that the COPD vs non-COPD distinction is best captured by gene and protein-level signalling pathways as opposed to metabolic pathways, but it may also be due to the lower metabolite coverage, and smaller set of pathways accessible using these molecules (Table 1, Table A in S1 Supporting Information).

We then investigated the pathways ranked highly by MB-VIP across all latent variables. Pathway importances can be queried at an individual omics level (Fig 5C), or at a multi-omics level with VIP normalised across all views (Fig 5D). The same is also possible at the individual latent variable level, and as superscores are orthogonal, each latent variable contains a different combination of pathways contributing to the prediction of $y$. $p$-values for the MB-VIP statistic were computed empirically using permutation testing (see Methods). In Fig 5C we observe that the metabolic pathways implicated in COPD pathology relate broadly to fatty acid metabolism, including carnitine metabolism, as well as central carbon metabolism [46]. The transcriptomics layer also highlighted the importance of glycogenolysis (glycogen breakdown), which alongside alterations in lipid metabolism have been found to be implicated in severe COPD, where there is an increased dependence on glucose for energy production due to impaired lipolysis, and hence an increased rate of glycolysis [47]. Carnitine metabolism was one of the top ranked (metabolic) pathways overall, with Fig 5D showing its significance was driven by the metabolomics layer ($p$ = 0.003). The 'Carnitine metabolism' pathway is composed of both metabolites and proteins, of which there was also sufficient coverage in the transcriptomics data to produce ssPA scores for this pathway. In the transcriptomics data however, this pathway was not significant ($p$ = 0.55); this demonstrates the benefit of multi-omics modelling to gain a broader perspective of the molecular basis of disease. Fatty acid metabolism has been shown to be part of a metabolic reprogramming that occurs in respiratory disease including COPD[48,49]. In COPD specifically, impairments in the carnitine shuttle system in the mitochondria (preventing long-chain fatty acids from being transported into the mitochondria) have been shown to result in lipotoxicity within the cell cytosol [50–52]. Conversely, 'Surfactant metabolism', which did not have sufficient coverage to be included in the model in the metabolomics view, but was found relevant in the proteomics data

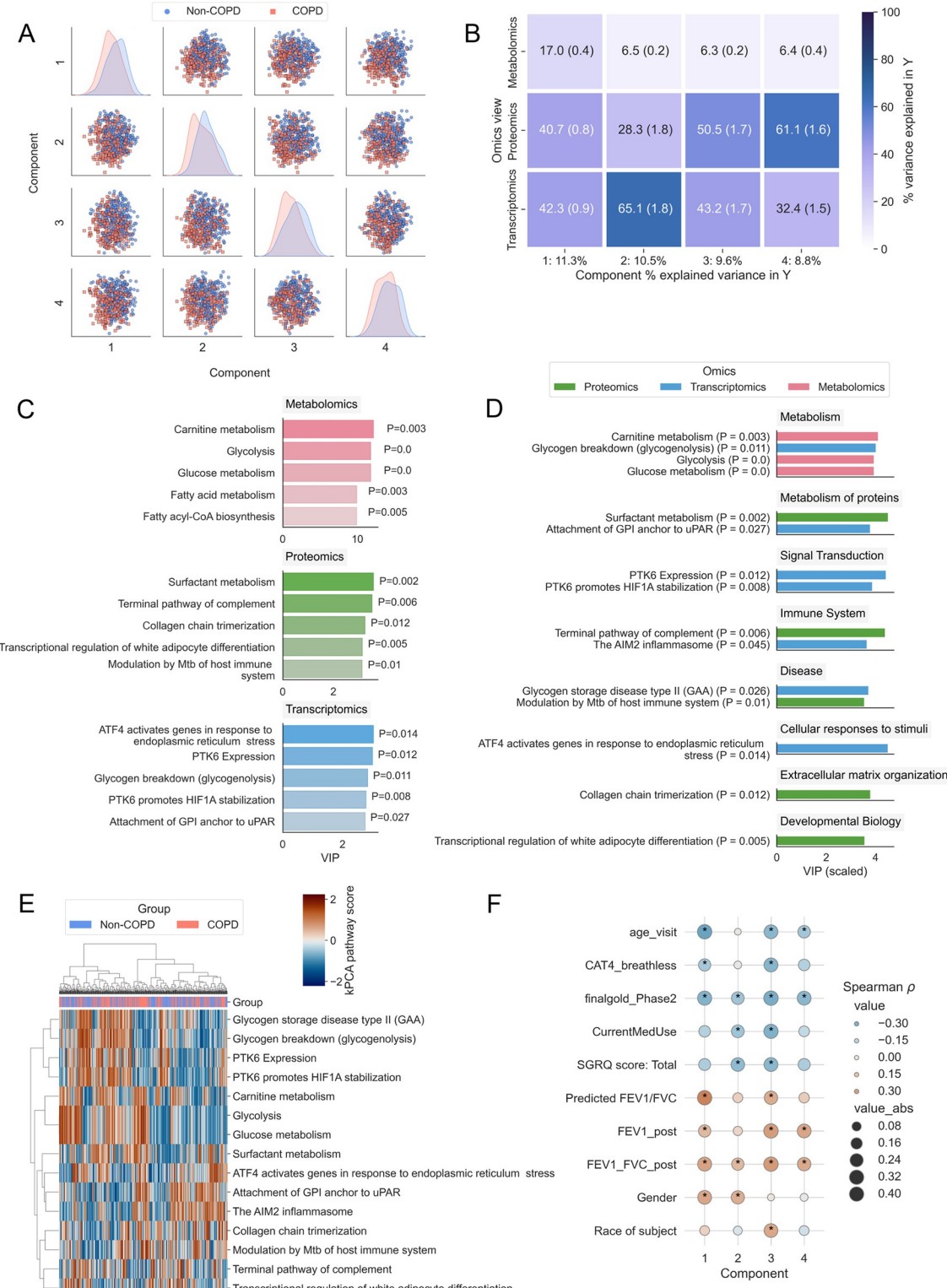

**Fig 5. PathIntegrate Multi-View applied to COPDgene multi-omics data.** A. Superscores plot based on multi-omics (metabolomics, proteomics, and transcriptomics) pathways across four latent variables. B. Omics view importances across latent variables. Values represent mean and SEM across 100 bootstrap samples. C. Top five pathways per omics block. D. Top 15 pathways across omics blocks categorised by Reactome parent pathway. E. kPCA ssPA scores from top 15 pathways used to cluster samples using Euclidean distance and Ward linkage. F. Heatmap showing Spearman correlation between superscores across four latent

variables and clinical metadata. Asterisks indicate Bonferroni p-value ≤ 0.05. Definitions of clinical variables are in Table B in S1 Supporting Information.

($p$ = 0.002), is an important process by which phospholipid surfactants are produced by the alveoli to ensure optimal lung function [53]. The surfactant lipidome has been found to be significantly different in COPD patients compared to healthy controls and is a potential therapeutic target [53]. Finally, several relevant proteomics and transcriptomics pathways involved focus on innate immune processes, highly important in the chronic inflammatory nature of COPD, such as inflammasome action ('The AIM2 Inflammasome') and the complement system ('Terminal Pathway of Complement'). The AIM2 inflammasome has recently been implicated in COPD pathogenesis, correlating with COPD severity and cigarette smoke exposure [54]. The full list of significant pathways is available in S1 File.

To demonstrate alternative visualisation strategies possible with PathIntegrate, we extracted the top 15 pathways across all omics ranked by MB-VIP from the Multi-View model and used the ssPA scores for these pathways to cluster the samples (Fig 5E). Hierarchical clustering showed two distinct clusters of pathways, one relating to metabolic processes such as central carbon and fatty acid metabolism, as well as hypoxia-associated signalling pathways ('PTK6 expression', 'PTK6 promotes HIF1A stabilization'), and the other consisting of processes involved in the innate immune response ('The AIM2 inflammasome', 'Terminal pathway of complement').

Further interpretation of the model can be gained by examining the correlation between the superscores for each latent variable and clinical metadata, enabling investigation of the relationship between clinical features and pathways (Fig 5F). For example, we found pathways in latent variables 1, 3, and 4 to be significantly associated with age, whereas pathways in latent variable 3 were significantly associated with the race of subjects.

Finally, to check that the pathway-based modelling approach does not appreciably degrade prediction performance, we examined the performance of PathIntegrate Multi-View versus a molecular-level MB-PLS model using the COPDgene dataset (Table 2). In the case of predicting COPD using plasma multi-omics data (metabolomics, proteomics, and transcriptomics), for example, the pathway level model achieved an average AUC of 0.70 (±0.02), and the molecular level model also achieved an average AUC of 0.70 (±0.02) when using all molecules available (inc. those not mapping to pathways), but required more latent variables to do so (4 vs. 6), resulting in a more complex model (Table 2).

Visualisation of high-dimensional omics data in the context of many hundreds of pathways remains a challenge. Alongside typical graphical outputs from the model, the PathIntegrate

**Table 1. Number of Reactome/KEGG pathways accessible in COPDgene and COVID-19 multi-omics datasets.**

| Dataset | Number of Reactome pathways accessible (≥ 2 molecules mapping) | Number of KEGG pathways accessible (≥ 2 molecules mapping) |
|---|---|---|
| COPDgene metabolomics | 202 | 125 |
| COPDgene proteomics | 1396 | 291 |
| COPDgene transcriptomics | 1902 | 341 |
| COVID-19 metabolomics | 169 | 122 |
| COVID-19 proteomics | 599 | 217 |

**Table 2. Performance comparison of PathIntegrate Multi-View using pathways versus using the molecular-level COPDgene dataset (mean AUC and 95% CI, as well as the number of latent variables (LV) used).** In both pathway and molecular-level scenarios the model was used to predict binary COPD status. The molecular-level model was fit both with all molecules available in the datasets, as well as only those mapping to pathways. AUC values are averaged across 5-times repeated 5-fold cross validation.

| | All omics | Metabolomics and proteomics | Metabolomics and transcriptomics | Transcriptomics and proteomics | Metabolomics | Proteomics | Transcriptomics |
|---|---|---|---|---|---|---|---|
| AUC (pathway) | **0.70** (0.67, 0.72) (4 LV) | **0.67** (0.66, 0.69) (3 LV) | **0.69** (0.67, 0.71) (3 LV) | **0.68** (0.66, 0.70) (4 LV) | **0.63** (0.61, 0.64) (1 LV) | **0.67** (0.66, 0.68) (3 LV) | **0.65** (0.63, 0.66) (3 LV) |
| AUC (molecular) | **0.70** (0.69, 0.72) (6 LV) | **0.71** (0.70, 0.72) (2 LV) | **0.70** (0.68, 0.71) (6 LV) | **0.71** (0.70, 0.73) (7 LV) | **0.66** (0.65, 0.69) (2 LV) | **0.72** (0.71, 0.74) (3 LV) | **0.68** (0.66, 0.69) (5 LV) |
| AUC (molecular–only those mapping to pathways) | **0.72** (0.70, 0.74) (7 LV) | **0.72** (0.70, 0.74) (2 LV) | **0.67** (0.66, 0.69) (6 LV) | **0.70** (0.69, 0.72) (6 LV) | **0.68** (0.67, 0.7) (2 LV) | **0.71** (0.70, 0.73) (3 LV) | **0.66** (0.64, 0.68) (7 LV) |

package provides an interactive network explorer app designed to visualise the results of PathIntegrate models on the Reactome pathway hierarchy graph (Fig L in S1 Supporting Information). Nodes in the network represent pathways and edges represent parent-child relationships between them as part of a directed acyclic graph (DAG). Nodes can be coloured by feature importance in the PathIntegrate model, so that users can intuitively visualise important pathways and their relationships to other areas of the pathway network. Various hierarchical and force-directed layouts are available, and images can be exported for further annotation and customisation. Fig 6A shows a global overview of the Reactome pathway network based on coverage of the COPDgene dataset (full pathway hierarchy legend shown in Fig M in S1 Supporting Information). We coloured nodes by MB-VIP $p$-values in Fig 6B to identify important pathways linked to COPD, as well as other pathways which may be affected by proximity in the network. Fig 6B highlights the 'Carnitine metabolism' pathway ($p \leq 0.05$), as well as other pathways which may not have reached statistical significance but may be of interest such as 'Arachidonic acid metabolism, or 'Mitochondrial fatty acid beta oxidation' [48,55]. Encouragingly, related pathways in the close neighbourhood of 'Carnitine metabolism' have lower $p$-values than those further from it.

Taken together, these results demonstrate how PathIntegrate Multi-View can be used to investigate various aspects of pathway regulation associated with a specific phenotype. COPD-associated pathways can be explored both within omics (individual views) and across omics (global view), and superscores of the latent variables can be used to identify correlations between pathways and other data, e.g. clinical measurements. The contribution of each omics to the prediction can be easily obtained from the Multi-View model, which obtains a lower-dimensional representation of the data that maximises covariances between omics view blocks and the $y$ outcome, but also keeps data blocks separate in order to retain this level of granularity.

## PathIntegrate Single-View applied to COVID-19 multi-omics data

We applied PathIntegrate Single-View to data from a multi-omics study of COVID-19 severity [56] to understand pathways driving the transition from mild to moderate/severe COVID-19 pathogenesis. Proteomics and LC-MS metabolomics data were integrated using PathIntegrate Single-View, in which the concatenated omics data were transformed to multi-omics ssPA scores using the SVD method [37] and a random forest model was applied to the resulting Reactome pathway score matrix.

An advantage of the Single-View model is that it computes each pathway score based on multi-omics data, providing a broader coverage of pathways by doing so (Fig 7A). Multi-

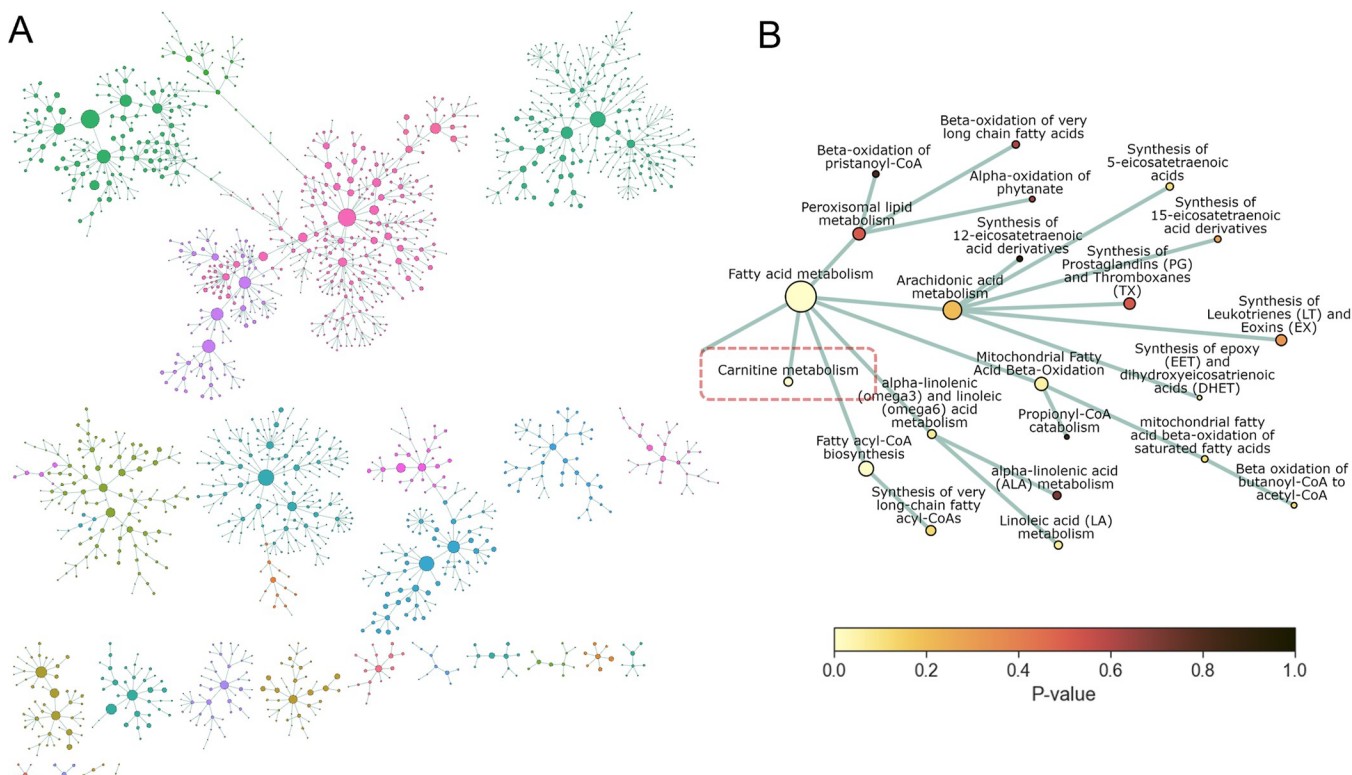

**Fig 6. Network visualisation with PathIntegrate interactive network explorer.** PathIntegrate Multi-View was applied to COPDgene multi-omics data. A. Multi-omics network view of global Reactome hierarchy DAG. Only pathways with sufficient coverage (≥ 2 molecules per pathway) are shown as nodes. Edges represent parent-child relationships between pathways as defined by Reactome. Nodes are coloured by Reactome superpathway membership. Node size corresponds to pathway coverage. B. Network view of 'Carnitine metabolism' pathway (zoomed-in susbset of (A)) and close neighbourhood within the Reactome pathway hierarchy. Nodes are coloured by p-values obtained from PathIntegrate Multi-View model.

omics pathways had a greater mean pathway coverage (number of molecules in the data mapping to each pathway, mean: 6.39 versus 6.21 and 4.86 for proteomics and metabolomics separately). This enabled more pathways to be included as they contained enough molecules to meet the minimum filtering threshold (732 pathways versus a maximum of 599 and 169 for proteomics and metabolomics separately, a total of 701 unique pathways); we used a liberal threshold of ≥2 molecules per pathway (Fig 7B).

We found the PathIntegrate Single-View model to perform similarly in terms of classification AUC on the unseen test set (AUC 0.95) compared to the concatenated molecular level omics data (AUC: 0.98), suggesting that in this case pathway-level modelling can aid interpretation without substantial loss of prediction performance. We next inspected the important multi-omics pathway features using random forest recursive feature elimination, which identified 20 of the most informative pathways (Fig 7C). Within this set, there are several immune-related processes known to be implicated in COVID-19 severity such as 'Interleukin-5 and interleukin-13 signalling' [57,58] and 'Caspase activation via death receptors in the presence of ligand' [58].

Finally, if certain ssPA methods are used (e.g. SVD [37]), it is possible to obtain information on how individual molecules contribute to the formation of the overall multi-omics pathway score. As we used SVD scores in this model, we can use the loadings on principal component 1 as the importance of each molecule in the pathway score (Fig 7D). In Fig 7D, which shows the molecular-level importance for the 'ADORA2B mediated anti-inflammatory cytokines

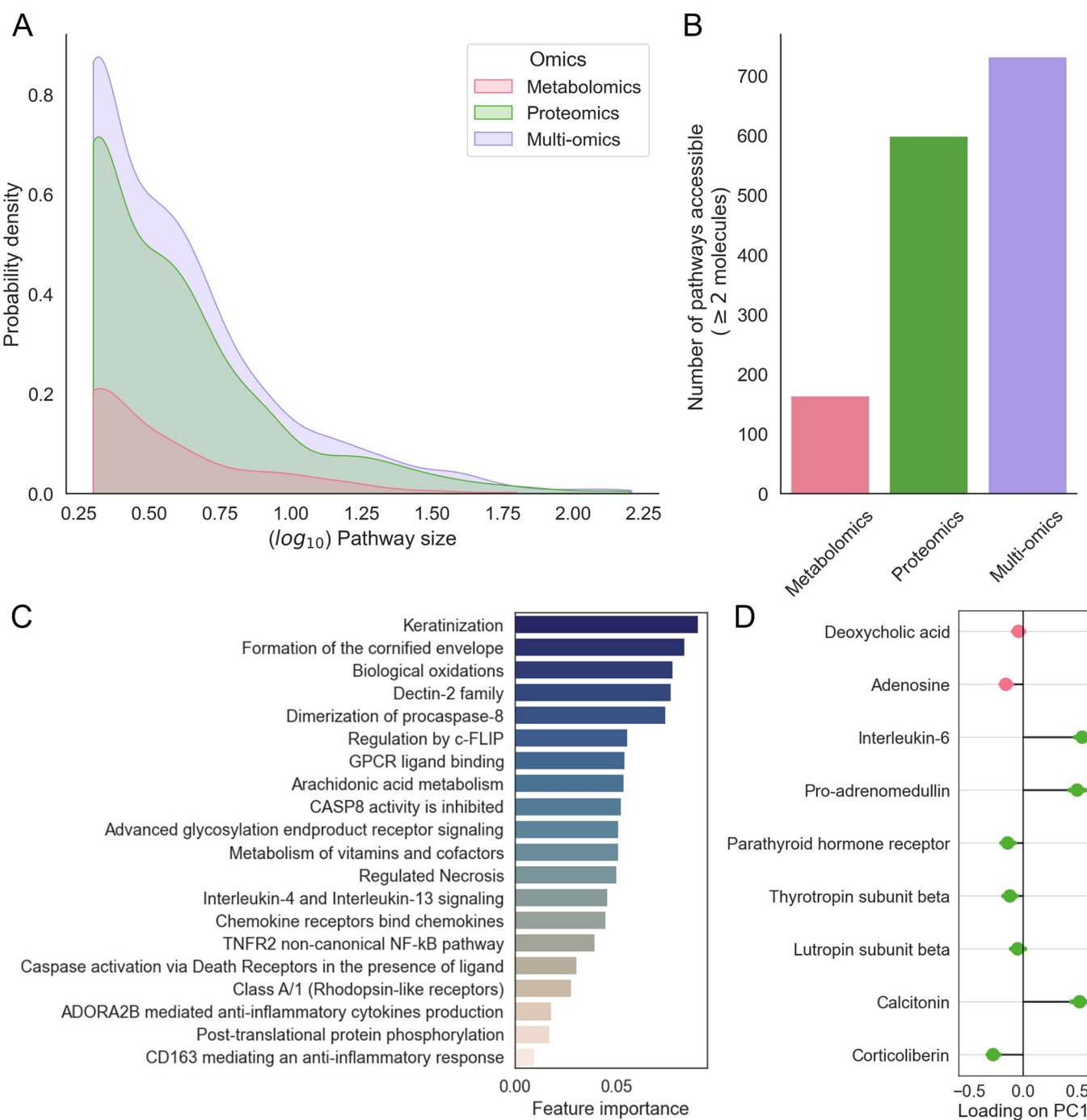

**Fig 7. PathIntegrate Single-View applied to COVID-19 multi-omics data.** A. Kernel density distribution of $log_{10}$ pathway sizes in the COVID dataset per omics view. Pathway size refers to the number of molecules annotated to each pathway present in the COVID datasets. B. Number of pathways with sufficient coverage in the COVID dataset in each omics view. C. Multi-omics pathway features identified using recursive feature elimination from the PathIntegrate Single-View random forest model, ranked by Gini importance. D. Molecular level importances derived from the 'ADORA2B mediated anti-inflammatory cytokines production' (R-HSA-9660821) SVD pathway scores. Datapoints represent mean and standard deviation of loadings of each molecule on PC1 across 200 bootstrap samples.

production' pathway as an example, we observe that metabolites deoxycholic acid and adenosine are correlated with four proteins, all with negative loadings on PC1, while three proteins: interleukin-6 (IL-6) and the hormones pro-adrenomedullin and calcitonin had positive loadings with greater magnitudes. In chronic COVID-19, elevated levels of both IL-6 and adenosine have been observed, with IL-6 contributing to the proinflammatory 'cytokine storm' and adenosine being considered as a potential therapeutic for severe cases due to its anti-inflammatory effects [59]. Such investigations can help researchers pinpoint the specific molecules contributing most to pathway scores, reducing the number of molecules required in developing biomarker assays, as well as providing understanding of how molecules from different omics correlate in the latent space.

## Discussion

This study contributes a new approach to the rapidly growing body of multi-omics integration methods [3,5,6,27], specifically by providing insights into the use of pathways as a basis for interpretable predictive modelling of multi-omics data, and by introducing the PathIntegrate framework for doing so. The use of pathways for modelling omics data is a promising avenue of research, with several studies highlighting its potential in recent years [27,60]. However, there is limited research available on the use of pathways for multi-omics integration, or evaluation of the performance of pathway-based versus molecular-level integration models. Here, we have introduced the PathIntegrate Multi-View and Single-View modelling frameworks for multi-omics pathway-based integration and evaluated their performance using semi-synthetic and experimental data.

To demonstrate the ability of pathway transformation to increase statistical power by combining correlated molecular signals, we applied a series of univariate tests to evaluate the ability to detect pathway or molecular level enrichment across various effect sizes. At lower effect sizes, we found that the univariate tests could recover more pathway-level signals than molecular signals, demonstrating the benefit of pathway transformation of multi-omics data, which often have low effect sizes, particularly in heterogeneous clinical studies, and especially those where a phenotype is not well defined. Additionally, pathway transformation naturally reduces the number of tests required, thereby reducing the multiple testing correction burden. This motivated our development of PathIntegrate, which uses ssPA pathway transformation as a basis for pathway-based multi-omics integration.

We compared PathIntegrate to DIABLO [11], a highly-cited multi-omics integration tool, which uses a similar underlying multi-view model as PathIntegrate Multi-View. We found PathIntegrate methods to perform similarly to DIABLO (when using pathway score matrices as input). Overall, however, we wish to emphasise the benefit of using pathway-transformed data as input to multivariate models and show that even using a different predictive model (DIABLO RGCCA vs PathIntegrate Multi-View MB-PLS) similar results can be obtained. We compared PathIntegrate Multi-View to a molecular level MB-PLS model and demonstrated the ability of the pathway-based model to classify samples with improved AUC across effect sizes. A full comparison of PathIntegrate, a pathway-based predictive model, to conventional pathway analysis approaches, such as ORA [25], GSEA [26], or integrated pathway analysis e.g. MultiGSEA [19] and IMPaLA [16] is beyond the scope of the present work. This is because pathway-based predictive models leverage multivariate modelling to identify pathways most associated with an outcome, whereas conventional pathway analysis methods typically test pathways in a univariate and non-predictive manner. Although the question of 'which pathways are perturbed in a phenotype?' is similar in both approaches, the way results are derived and the differences in outputs would render a direct comparison challenging, yet an interesting avenue for future research.

We applied PathIntegrate to two datasets: COPDgene and COVID19 multi-omics. Both case studies highlighted the benefits of pathway-based modelling for integration, interpretation, and visualisation of multi-omics data. In terms of predictive performance, in both case studies, as expected, PathIntegrate performed similarly to the molecular level counterpart. Pathway coverage, the proportion of molecules in a pathway which can be observed in the data, or pathway annotation, the proportion of known documented biomolecules annotated to pathways in databases are both inherent bottlenecks of pathway-based analyses. These issues particularly affect certain datatypes such as metabolomics, where even multiple assays are not enough to provide high coverage of the metabolic pathway network [44]. Despite this, in the COVID-19 case study where 314 metabolites were annotated to ChEBI identifiers, and 456 proteins to UniProt identifiers, the PathIntegrate Single-View model based on 732 multi-omics pathway scores was still able to achieve an AUC of 0.95 in predicting COVID-19 severity based on the pathway coverage provided by these molecules. In the COPDgene case study, we compared the predictive performance of the PathIntegrate Multi-View model to its molecular level counterpart, including molecules not annotated to pathways. We did not find including un-annotated molecules lead to significant changes in performance in this case, however the reliance on current knowledge is a widely accepted limitation of pathway-based methods, which can be expected to improve as annotations become more complete. Meanwhile, future work should focus on how un-annotated molecules could be incorporated into pathway-based models, to minimise information loss.

Another important consideration is pathway database choice, as pathway definitions can differ greatly between databases, as well as the level of overlap between pathways and possible hierarchical structure [44,61–63]. As expected, we found PathIntegrate to exhibit minor changes in predictive performance based on the database used. While the pathway transformation step in PathIntegrate can result in transformed omics datasets having more homogenous numbers of pathway features, this is not always the case, and depends on several factors, including the pathway resource used, the coverage of the assay technology, as well as the underlying biology (e.g. there are inherently fewer known metabolic pathways than signalling pathways).

Although PathIntegrate Multi-View uses an MB-PLS model and Single-View uses any Sci-KitLearn-compatible predictive model (e.g., random forest), we endeavour to provide readers with a general framework for pathway-based multi-omics integration which they can build upon to complement their experimental design or analysis goals. For example, if prediction of a phenotype with high accuracy is a desired outcome, a deep feed forward neural network could be applied within the Single-View framework, to classify samples based on pathways. Model interpretability can also be further enhanced by customising the model inputs, such as using bespoke pathway sets or ontologies to generate the pathway score input layer. For example, in PathIntegrate Multi-View, an additional omics block could be added composed of lipidomics data, and pathway scores could be computed using the LipidMaps[64] classification system to reflect enrichment patterns of lipid subclasses. Note that in this work we focused on supervised pathway-based integration models; however similar frameworks using unsupervised methods are also feasible and may be explored further. We decided to focus on supervised methods as firstly an outcome is directly modelled and there is less risk of confounding variation obscuring the interpretation, and secondly, users can evaluate model performance in a straightforward manner by examining prediction accuracy.

Both PathIntegrate Single-View and Multi-View are designed to handle multiple omics views. In this work we have demonstrated the use of two or three omics views, however both models can accommodate further (3+) omics views as long as they contain continuous measurements (rather than binary e.g., genomics data) and the features can be mapped to pathway

identifiers, enabling the pathway-transformation stage to be performed. Data blocks from the same omics type e.g. metabolomics but profiled on different biofluids or tissues can also be integrated using PathIntegrate, to understand how pathways in different biological matrices contribute to the phenotype. Although the focus of this work was on pathway-based models, both PathIntegrate models can be made hybrid in the sense that both pathway-transformed omics data and other data e.g., clinical metadata, genomics data, metagenomic data, etc., can be integrated alongside one another.

PathIntegrate is unique in its specific support for metabolomics in multi-omics studies, which is often omitted by other integration methods. Metabolomics is becoming frequently profiled alongside gene-based omics, providing researchers with an essential snapshot on the biochemical activities of small-molecules [1,65]. Metabolomics data differs considerably from gene-based omics in several ways including the molecular identifiers used, assay coverage of the metabolome, and annotation uncertainties. PathIntegrate users can download the latest release of Reactome pathways via the sspa Python package and obtain a merged multi-omics pathway database object composed of protein (UniProt), gene (ENSEMBL), and metabolite identifiers (ChEBI) to enable integration of these distinct omics in a straightforward manner.

Our study shares several limitations with other pathway-analysis and multi-omics integration studies, a key drawback being the lack of appropriate benchmarking data. Ideally, a benchmarking dataset would contain two or more high-quality omics views, a large sample size (n $\geq$ 1000), and known biological signals at the molecular and pathway level validated by laboratory experiments. Without access to such data, we employed the semi-synthetic simulation strategy to artificially introduce known molecular and pathway-level signals into a real experimental dataset. As described in our previous work [33], this approach allows the simulation to retain important characteristics of real data such as the underlying statistical distributions, correlations, and covariances between molecules and pathways. It also enabled us to vary the effect size of pathway signals, which we based on the effect sizes ($log_2$ fold changes) detected in the experimental datasets used. Despite these efforts, it remains a challenge to compare molecular vs. pathway-level models, as it is unknown how many molecules in a pathway are differentially abundant at any one time, and pathway definitions and sizes vary between databases [44,63,66,67].

In common with many other statistical integration approaches, PathIntegrate requires all input omics to be measured on the same individuals. This means samples from individuals without data on all omics will have to be discarded, as PathIntegrate currently does not support entire rows of missing data. Some models can accommodate sparse data where values are missing at random [68], including MB-PLS (using NIPALS algorithm [69]) and probabilistic frameworks such as MOFA [13]. However, further work is required to develop multi-omics integration methods that can handle samples with one or more omics missing [70] (missing not at random [68]). Additionally, using pathway-based models may aid in the robust imputation of data, by helping to capture biological rather than technical variation.

## Conclusion

As knowledge of biological pathways continues to evolve and pathway databases develop alongside this, we anticipate that pathway-based models such as PathIntegrate will become a valuable way of interpreting complex multi-omics datasets. This work contributes to our understanding of such models, by evaluating the effectiveness of using pathways for multi-omics integration, as well as introducing the PathIntegrate modelling framework. PathIntegrate provides a novel solution to the challenge of integrating heterogeneous omics datasets, by using pathway-transformation to bring omics to a common basis, followed by state-of-the-

art supervised modelling. The PathIntegrate framework presented here and accompanying Python package will provide a useful resource to the research community, streamlining the analysis of multi-omics data with the aim of providing an interpretable, integrated set of results at the pathway level.

## Methods

### Datasets

**COPDgene data.** We integrated COPDgene Phase 2 (~5 years after baseline) plasma metabolomics (Metabolon UHPLC-MS/MS), plasma proteomics (SOMAscan 1.3k assay), and bulk whole blood transcriptomics data (Illumina HiSeq2000) from 522 samples which had data for all three omics. As detailed in Regan et al., 2010 [45]: COPD was defined using spirometric evidence of airflow obstruction [post-bronchodilator forced expiratory volume at one second (FEV1)/forced vital capacity (FVC) $\geq$0.70], as well as a GOLD score of 1–4. The subcohort comprised 273 COPD samples (GOLD 1–4) and 249 non-COPD samples (GOLD 0) from smokers. Full details of the multi-omics datasets and pre-processing are available in the original article [21]. We also obtained clinical data for samples, including COPD phenotypes and demographic variables. Clinical data was filtered to include 260 variables measured in all 522 samples of the sub-cohort.

**COVID-19 data.** The publicly available COVID-19 multi-omics dataset was obtained from Su et al. 2020 [56]. Full details of the multi-omics datasets and pre-processing are available in the original article [56]. We integrated plasma metabolomics (Metabolon UHPLC-MS/MS) and proteomics (Olink) datasets with matched samples, of which 45 samples had 'mild' COVID (WHO status 1–2), and 82 had 'moderate-severe' COVID19 (WHO status 3–7), totalling 127 samples.

**Multi-omics data pre-processing and quality control.** All multi-omics datasets were subject to quality control and pre-processing as detailed in the original articles [45,56]. Metabolomics, proteomics, and transcriptomics abundances were $log_2$ transformed followed by unit-variance scaling. Missing values were imputed using the singular-value decomposition approach implemented in the fancyimpute Python package. In the transcriptomics data, low-variance genes (below 25th percentile) were filtered out. Table 3 shows the number of molecules in each omics remaining after identifier mapping and quality control which were used in all analyses.

*Identifier mapping.* Identifier harmonisation of both the COPDgene and COVID metabolite datasets was performed via the sspa package identifier conversion utility via the MetoboAnalyst [71] API (https://www.metaboanalyst.ca/docs/APIs.xhtml.) HMDB metabolite identifiers provided with the datasets were converted to ChEBI (for Reactome)/KEGG compound (for KEGG) identifiers.

COPDgene and COVID-19 proteomic data was provided with UniProt identifiers which directly map to Reactome pathways. KEGG gene IDs were obtained using the UniProt ID matching tool (https://www.uniprot.org/id-mapping). COPDgene transcriptomics data was provided with ENSEMBL IDs which directly map to Reactome pathways.

**Table 3. Number of molecules in each omics in COPDgene and COVID-19 datasets after processing and identifier mapping.**

| Dataset | Total number of samples | Number of metabolite features (mapping to ChEBI) | Number of protein features (mapping to UniProt) | Number of transcript features (mapping to ENSEMBL) |
|---|---|---|---|---|
| COPDgene | 522 | 513 | 1305 | 14441 |
| COVID-19 | 127 | 314 | 456 | NA |

## Pathways

PathIntegrate Single-View and Multi-View models both make use of a single, merged set of multi-omics pathways as input. Each pathway contains either a set of molecules from different omics (metabolites (ChEBI), proteins (Uniprot), and genes (ENSEMBL)), or only molecules from a single omics, depending on the pathway definition. The PathIntegrate package enables download of multi-omics pathway sets (via sspa) from Reactome, providing a text file of the latest version for various supported organisms in standard GMT file format. PathIntegrate is also flexible to the input pathway set and is not restricted to those provided via the package. Any pathway set in GMT file format can be used as input, where each row represents a pathway, and each pathway set is described by a name, a description, and its constituent molecules (see Broad Institute website for further details on GMT format: https://software.broadinstitute.org/cancer/software/gsea/wiki/index.php/Data_formats#GMT:_Gene_Matrix_Transposed_file_format_.28.2A.gmt.29).

In this work, Reactome human version 83 and KEGG human version 105 were used. Table 1 shows the number of pathways from each omics in the COPDgene/COVID-19 datasets accessible using the molecules profiled in each dataset ($\geq 2$ per pathway).

## Semi-synthetic multi-omics data generation

To benchmark our methods, we applied the semi-synthetic simulation approach detailed in Wieder et al., 2022 [33] to insert artificial biological signals into existing multi-omics data. This approach involves creating simulated datasets based on experimental data, with the assumption that doing so will preserve the complex biological signals and statistical distributions within the data, and more accurately reflect a real scenario as opposed to approaches based on sampling from parametric distributions. Various experimental designs can be simulated using this approach, but here we opt for a simple case-control design in which we add the artificial signal only to molecules in the 'case' group. By adding the same effect size to the abundances of all molecules within a pathway (detailed below), this approach emphasises realism (by preserving the covariance structure of the original omics data) without being overly complex.

The input data is a series of $\log_2$ transformed abundance matrices for the $k$ omics types $X_k = [x_1, x_2, \ldots, x_{M_k}]$, each of size ($N \times M_k$), and a set of $N$ outcome labels $y_i, i = 1, \ldots, N$. The approach is as follows for each realisation of the semi-synthetic data:

1. Randomly shuffle outcome labels $y_i$. This results in a new 'control' group $C$ and a new 'case' group $D$ of the same class sizes as the original dataset. The shuffling ensures any biological effects correlated to the outcome are removed but preserves existing covariances between molecules.

2. Add a constant $\alpha$ corresponding to desired effect size (e.g. $\log_2$ FC = 0.5) to specified target molecules only in samples in the new 'case' group $i \in D$, simulating increased abundance of those molecules associated with the outcome (Eq 1).

$$X_{i,j} \rightarrow X_{i,j}, \quad i \in C$$

$$X_{i,j} \rightarrow X_{i,j} + \alpha, \quad i \in D \qquad\qquad \text{Eq1}$$

In this work we increase the abundance of all molecules in a single target pathway at each realisation, at the same effect size. By adding a constant to $\log_2$ scale data this simulates a multiplicative fold change in the original data.

In this work, we enriched all molecules in one randomly selected (Reactome/KEGG) 'target' pathway $p_i$ at a time, at varying effect sizes. Here, effect size refers to the $log_2$ fold change of a molecule. We enriched the known target pathway by effect sizes of 0–1 in the COPDgene dataset and 0–3 in the COVID-19 dataset, based on fold changes observed in the original data (Fig A/Fig B in S1 Supporting Information). For performance evaluation purposes, we performed the semi-synthetic simulation approach using COPDgene and COVID-19 metabolomics and proteomics datasets. We performed the semi-synthetic simulation once for each target pathway in the Reactome/KEGG database that contained at least 3 molecules mapping to the input data (1290 and 298 realisations for Reactome and KEGG respectively for COPDgene data; 456 and 256 for COVID-19 data). For each target pathway we used a different random shuffling of outcome labels.

## Single-sample pathway analysis

Reactome human pathways (R83) and KEGG human pathways (R105) were downloaded using the sspa Python package v0.2.4 (https://github.com/cwieder/py-ssPA). The sspa package creates multi-omics pathways by merging proteins/genes and metabolites participating in the same pathway into a single multi-omics pathway.

Single-sample pathway analysis (ssPA) is an unsupervised method used to transform omics data matrices into pathway score matrices, where columns represent pathways rather than individual molecules (Fig 1). Importantly, all omics data input to ssPA must be standardised. Throughout this work and in the ssPA Python package, unit variance scaling is used, where the mean of each feature is set to 0 and the standard deviation is set to 1. ssPA begins by using the P pathways $P = \{p_1, p_2, \ldots, p_P\}$ passing minimum coverage criteria for the dataset (an integer defined by the user, default 2 molecules per pathway). The $i$'th pathway $p_i$ is composed of $L_i$ molecules (e.g. proteins), $p_i = \{m_1, m_2, \ldots, m_{L_i}\}$. ssPA is performed to provide pathway 'activity scores' for each sample, reflecting an estimate of the enrichment of each pathway in each individual sample.

One of the most popular categories of ssPA methods is that based on dimensionality reduction, specifically PCA. In the original PLAGE (referred throughout this work as 'SVD') method by Tomfohr et al. [37], singular value decomposition is performed on the omics abundance matrix retaining only the $L_i$ columns (molecules) present in the $i$'th pathway. For each pathway, column vectors of abundance profiles belonging to molecules in pathway $p_i$ are concatenated to form a matrix $Z_i$ (Eq 2).

$$Z_i = [x_{m_1}, x_{m_2}, \ldots, x_{L_i}]$$                    Eq2

Then, the first right singular vector (first principal component score) is used to represent the pathway 'activity' scores $a_i$ (size $N \times 1$) for the $i$'th pathway. Pathway score vectors for each pathway are combined to produce a sample-by-pathway matrix $A = [a_1, a_2, \ldots, a_P]$. The kPCA method we proposed in [33] uses a very similar approach, instead applying kernel PCA with a radial basis function kernel and using the scores for principal component 1 to reflect pathway activities. Full details of how ssPA is performed are available in [33,36,37]. In this work we used the kPCA method [33] in the benchmarking section and COPDgene application, and the SVD method (PLAGE) [37] in the COVID-19 application section. The sspa package functions sspa_KPCA and sspa_SVD were used to generate pathway score matrices used in both PathIntegrate Multi-View and Single-View.

## Supervised modelling frameworks

**PathIntegrate Single-View.**   PathIntegrate Single-View is a predictive model applied to a single data matrix of multi-omics ssPA scores (Fig 3). Conceptually it is simpler than PathIntegrate Multi-View due to the input being a single pathway-level matrix rather than multiple pathway-level matrices. Note that both models integrate multi-omics data; the "Single-View" and "Multi-View" refer to the machine learning framework used to effect this integration.

The first step of PathIntegrate Single-View involves computing ssPA scores at the multi-omics level, using multi-omics pathway sets (i.e. pathways $p_i = \{m_1, m_2, \ldots, m_{M_i}\}$ where the $m_i$ represent genes, metabolites, and proteins present in the omics data). All input omics data matrices are unit-variance scaled. ssPA is performed on the multi-omics abundance matrices $Z_i$ using any sspa algorithm implemented in the sspa Python package to form the pathway scores matrix $A$ of size ($N \times P$).

The second step of PathIntegrate Single-View applies a predictive model to the multi-omics ssPA score matrix $A$ to predict an outcome variable $\hat{y}$ (Eq 3).

$$\hat{y} = f(A; \theta) \qquad\qquad\qquad \text{Eq3}$$

where $\theta$ represents the parameters of the predictive model $f$. There is a single predictor matrix, hence the term 'Single-View'. The user can apply a variety of models (any of those available in SciKitLearn are compatible with the PathIntegrate python package), including random forest, PLS regression, support vector machine, etc. Important pathways are determined using feature importance metrics specific to the predictive model used (e.g. Gini impurity for random forests or VIP for PLS regression). In this work to demonstrate PathIntegrate Single-View, we applied a PLS-DA model in the performance evaluation section, and a Random Forest model in the COVID-19 case study.

**PathIntegrate Multi-View.**   PathIntegrate Multi-View leverages multi-table integration approaches to build a predictive model based on multiple, separate ssPA score matrices from each omics view (Fig 3). There are several ($k>1$) predictor matrices here, hence the term 'Multi-View'. In this work we used a multi-block partial least squares (MB-PLS) model due to its ability to model multiple data blocks (omics views) in relation to a response variable $y$. However, any multi-view supervised machine learning technique could be used within the same framework. The MB-PLS model was implemented using the mbpls Python package [40] using the NIPALS algorithm. Again, all input omics data matrices are unit-variance scaled. As with PathIntegrate Single-View, users can apply any ssPA algorithm implemented in the sspa package to perform the first step of Multi-View, transforming each omics abundance matrix $X_k$ of size ($N \times M_k$) into a pathway score matrix $A_k$ of size ($N \times P_k$). Then each pathway score matrix $A_k$ is modelled by MB-PLS, to predict an outcome variable. Important pathways are identified using the multi-block variable importance in projection (MB-VIP) statistic, detailed below (Eq 14). In this section we follow standard practice in describing how MB-PLS models an outcome $Y$ (which can be univariate or multivariate) using a several predictor matrices $X_k$, that, for PathIntegrate, correspond to the pathway scores matrices $A_k$.

(Single block) partial least squares (PLS) regression[72] is a supervised regression method designed to work well on high-dimensional and highly co-linear datasets due to its latent variable decomposition of both the predictor and response variables [41]. PLS performs a simultaneous projection of the unit variance scaled predictor matrix $X$, of size ($N \times J$), and a $Y$ response matrix, of size ($N \times H$), into a lower dimensional space (defined by latent variables, LVs) to maximise the covariance between the two projections (X scores, T and Y scores, U) (Eq 4). The low dimensional representation of the X data can be used to predict Y (Eq 6).

The PLS model as defined by Wold et al, 2001[72] is as follows:

The $X$ and $Y$ matrices are decomposed into scores and loadings such that:

$$X = TV^T + E$$

$$Y = UC^T + F \qquad \text{Eq4}$$

Here, $T$ and $U$ represent $X$ and $Y$ scores respectively, each of size ($N \times R$), for a model with $R$ latent variables. V, size ($J \times R$), represents $X$ loadings, C, size ($HxR$) represents Y weights, and E and F refer to residual matrices, sizes ($NxJ$) and ($NxH$) respectively, of independent and identically distributed (iid) noise. Matrix transpose is denoted by T.

The X scores, T are linear combinations of the original X variables multiplied by the X weights (coefficients):

$$T = XW^* \qquad \text{Eq5}$$

Where W*, size (JxR) denotes the weights matrix relating to the original variables, as opposed to W, size (JxR), which denotes the weights matrix computed from the deflated matrices (see Eq 8 below).

The X scores and Y weights are used to predict Y:

$$\hat{Y} = TC^T + G, \qquad \text{Eq6}$$

where $G$ is a further residual matrix.

PLS is performed sequentially, obtaining scores, loadings, and weights for each of $R$ latent variables. Importantly, the first pair of latent vectors $t$ and $u$ are selected such that the covariance between them is maximal:

$$(t, u) = \underset{(t, u)}{argmax} (cov(t, u)) \qquad \text{Eq7}$$

At each step, the model estimates, corresponding to the product of scores and loadings are subtracted from the current $X$ and $Y$ matrices (this step is termed deflation) so that the next set of latent vectors $r+1$ can be computed from a new $X_{r+1}$ and $Y_{r+1}$:

$$X_{r+1} = X_r - t_r v_r^T$$

$$Y_{r+1} = Y_r - t_r c_r^T, \qquad \text{Eq8}$$

with $X_1 = X$ and $Y_1 = Y$.

The optimal number of latent vectors is typically chosen using cross-validation approaches. Using Eq5, the prediction of Y can be re-written as:

$$\hat{Y} = XW^* C^T + G \qquad \text{Eq9}$$

(note the * does not denote multiplication) and thus the regression coefficients for each X variable are obtained using:

$$\beta = W^* C^T \qquad \text{Eq10}$$

The prediction of Y can finally be expressed in the form of a regression equation:

$$\hat{Y} = X\beta + G \qquad \text{Eq11}$$

Once the model is fit the scores, loadings, and weight matrices can be interpreted. Variable selection approaches for PLS methods include inspection of $\beta$ coefficients, as well as variable importance in projection (VIP) [73]. VIP is based on the PLS weights $W$ weighted by the proportion of Y explained in each latent variable (sum of squares) normalised by the total sum of squares across all LVs, and explains the influence of each $X$ feature on the model.

VIP for the $j^{\text{th}}$ variable is given by[74]:

$$\text{VIP}_j = \sqrt{\frac{J \cdot \sum_{r=1}^{R} (w_{rj}^2 \cdot SSY_r)}{SSY_{cum}}} \qquad \text{Eq12}$$

Here $J$ represents the number of features in $X$, $R$ is the number of latent variables (LVs), $w_{rj}$ is the weight of the $j^{th}$ feature in the $r^{th}$ LV, $SSY_r$ is the sum of squares of Y explained by the $r^{th}$ LV, and $SSY_{cum}$ is the cumulative sum of squares.

Often, variables with VIP <1 are discarded, as the average of sum of squares of VIP scores is equal to 1. However, a more reliable approach is to compute significance of the VIP values using empirical $p$-value computation, described below in section 'Feature importance'.

Multi-block PLS is an extension of PLS that allows multiple data blocks $\{X_1,\ldots,X_K\}$ as predictors [41]. The $k$'th $X$ predictor block and $Y$ response matrix can be decomposed as:

$$X_k = T_k V_k^{T} + E_k$$

$$Y = T_s C^T + \text{F} \qquad \text{Eq13}$$

where $T_S$ represents the X superscores.

In the multi-block PLS case, block scores for each $X$ block are combined to form superscores $T_s = [T_1, T_2,\ldots,T_K]$. The superscores are used to predict the response scores $U$, and also to deflate the $X_k$ blocks (if using the method proposed by Westerhuis and Coenegracht 1997), rendering the superscores orthogonal.

VIP can be computed for MB-PLS models by using the superscores $T_s$ across all blocks. In Eq 14, $SSY$ represents the proportion of $Y$ explained across all $X$ blocks, using the superscores $T_s$ rather than the scores $T$ as in Eq 12 for single-block VIP.

MB-VIP for the $j^{\text{th}}$ variable present in the $k^{\text{th}}$ block is given by:

$$\text{MB}-\text{VIP}_j = \sqrt{\frac{f \cdot \sum_{r=1}^{R} (w_{krj}^2 \cdot SSY_r)}{SSY_{cum}}} \qquad \text{Eq14}$$

where $f$ is the number of features across all blocks.

Similar to the original VIP definition, this MB-VIP metric satisfies the condition that the mean of the sum of squares of VIP scores per $X$ block equals 1.

$$\frac{SS(\text{MB}-\text{VIP})}{f \cdot \text{k}} = 1 \qquad \text{Eq15}$$

where SS(MB-VIP) represents the total sum of squares of the multi-block VIP values.

## Univariate detection of pathway versus molecular-level signals

Applying the semi-synthetic data generation approach detailed above, we generated semi-synthetic data for each pathway accessible in the COPDgene and COVID-19 metabolomics and proteomics datasets (1290 and 298 realisations for Reactome and KEGG respectively for COPDgene data; 456 and 256 for COVID-19 data) at a range of different effect sizes.

For the pathway-level simulation, we used the ssPA kPCA method to generate ssPA scores for each simulation. We then performed Mann Whitney U (WMU) tests to determine whether there was a significant difference in the pathway scores of the target enriched pathway in the simulated control and case groups. Bonferroni correction was used to obtain adjusted $p$-values.

For the molecular-level simulation, we performed MWU tests to determine whether there was a significant difference in each of the molecules in the target enriched pathway in the simulated control and case groups. Bonferroni correction was used to obtain adjusted $p$-values. To facilitate comparison with the pathway-level simulation, we used the Fisher method to combine $p$-values from each molecule in the target pathway. If at least 50% of molecules in the target pathway had a significant MWU test adjusted $p$-value ($\leq 0.05$), we combined them using Fisher's method to obtain the final $p$-value. If less than 50% of the molecules in the target pathway had an adjusted $p$-value of $\leq 0.05$, the combined $p$-value was set to 1.

## Performance evaluation

Unit-variance scaling, imputation, and ssPA transformation were performed separately on the test-train splits in order to avoid data leakage when evaluating the results of multivariate methods. Specifically, for ssPA, for each pathway the ssPA (PCA/kPCA) model is fit on the training data only and ssPA scores for the test data are derived from the fitted model. Hyperparameter tuning for the number of latent variables in the MBPLS/PLS models was performed using 5-fold nested cross-validation, and for all semi-synthetic datasets the optimal number of latent variables was 1 (as expected). Predictive performance was computed using 5 times repeated 5-fold cross-validation, and evaluated using the area under the Receiver Operator Characteristic (ROC) curve (AUC).

**DIABLO.** DIABLO requires tuning of a hyperparameter representing the design matrix, which regulates the strength of correlation maximised between each omics block. In this work we used DIABLO with a 'null' design (no correlation constraint) as in the original DIABLO paper [11], as our simulation setup was not designed to incorporate correlations between omics blocks.

**Detection of target pathway simulation.** For the target pathway simulation, we also used AUC to determine how well each method was able to detect the artificially enriched target pathway in each simulation realisation. To compute the AUC, the confusion matrix of true positives (TP), false positives (FP), true negatives (TN) and false negatives (FN) was defined as follows:

- TP: The target enriched pathway with $p_{adj} \leq 0.05$

- FP: A non-target pathway with $p_{adj} \leq 0.05$

- TN: A non-target pathway with $p_{adj} > 0.05$

- FN: The target enriched pathway with $p_{adj} > 0.05$

$p$-values for each pathway's feature importance (e.g. VIP/MB-VIP/DIABLO loading) were computed using permutation testing, see 'Feature importance' below.

When evaluating the ability of DIABLO to detect the enriched target pathway, we used two methods referred to as 'DIABLO pathway (loading)' and 'DIABLO pathway (sparse loading)'. 'DIABLO pathway (loading)' involved using the loadings in a non-penalised single component GCCA DIABLO model as the feature importances and calculating empirical p-values for these loadings as described below. 'DIABLO pathway (sparse loading)' involves using a sparse DIABLO rGCCA model with L1 penalty, where 5-fold, 5-times repeated cross-validation is used to

select the number of important features. Then, 25 bootstrap subsets of the data are obtained (each containing 400 samples in the COPDgene data or 60 in the COVID-19 data per class) and a sparse DIABLO model is fitted on each of these subsets. The test statistic for feature importance is defined as the proportion of the 25 bootstraps in which the pathway has a non-zero (sparse) loading. Intuitively, the target enriched pathway should be of high importance to the sparse model and therefore often appear in the significant features with a non-zero loading. Empirical $p$-values are also computed from the 'DIABLO pathway (sparse loading)' test statistic as described below.

## Feature importance

$p$-values for the significance of each feature (pathway) in the PathIntegrate models were computed empirically using a standard permutation test. We permuted class labels (Y) 10,000 times to obtain $p$-values with a resolution of 0.0001. $p$-values for each feature were calculated by counting the number of trials with test statistic (in this case VIP, MB-VIP, DIABLO loading, or non-zero proportion for DIABLO sparse) greater than or equal to the observed test statistic, and dividing this by 10,000. Multiple testing correction using the Benjamini Hochberg FDR method was then applied.

## PathIntegrate network explorer app

Plotly Dash Cytoscape v0.3.0 (https://github.com/plotly/dash-cytoscape) was used to create the PathIntegrate network explorer app within the PathIntegrate python package. The app can be launched from within the Python package and runs on a local host. NetworkX was used to create the base network based on the Reactome pathway hierarchy, which was downloaded from https://reactome.org/download/ (ReactomePathwaysRelation.txt). Nodes represent pathways and edges represent a parent-child relationship between them. The app takes as input a PathIntegrate Multi-View or Single-View model object and uses attributes such as feature importance to colour nodes.

## COPDgene case study

A PathIntegrate Multi-View model was fitted to COPDgene metabolomics, proteomics, and transcriptomics data, using multi-omics ssPA scores generated using the kPCA method. The optimal number of latent variables (4) used in the MBPLS model was identified using nested 5-fold cross-validation.

The superscores were correlated to 260 clinical metadata variables using Spearman correlation, and $p$-values were corrected for using Bonferroni correction. Absolute correlations $\geq 0.3$ and adjusted $p$-values $\leq 0.05$ were used to filter for significantly correlated metadata variables.

## COVID-19 case study

A PathIntegrate Single-View model was fitted to COVID-19 metabolomics and proteomics data, using multi-omics ssPA scores generated using the SVD (PLAGE) method, and employing a random forest for outcome prediction. The optimal hyperparameters for the SciKit-Learn RandomForestClassifier model selected via 5-fold cross-validatation were: n_estimators = 200, min_samples_split = 2, min_samples_leaf = 4, max_features = 'sqrt', max_depth = 10, bootstrap = True, oob_score = True.

**Identifying important pathways using PathIntegrate Single-View.** Random forest recursive feature elimination with 5-fold cross validation was used to identify the optimal

number of pathway features (20) for the Single-View model, implemented using the sklearn RFECV function.

**Identifying important molecules within a pathway.** For a pathway of interest, loadings on principal component 1 were used to represent the contribution of each molecule to the pathway scores across samples.

## Data and code availability

The COVID dataset is publicly available from Mendeley data (https://data.mendeley.com/datasets/tzydswhhb5/5) [56].

The COPDgene multi-omics data can be found at the following sources: Clinical Data and SOMAScan data are available through COPDGene (https://www.ncbi.nlm.nih.gov/gap/, ID: phs000179.v6.p2). RNA-Seq data is available through dbGaP (https://www.ncbi.nlm.nih.gov/gap/, ID: phs000765.v3.p2). Metabolon data is available at Metabolomics Workbench (https://www.metabolomicsworkbench.org/ ID: PR000907).

PathIntegrate is available via the open-source PathIntegrate Python package (www.github.com/cwieder/PathIntegrate). Tutorials and documentation for PathIntegrate can be found at https://cwieder.github.io/PathIntegrate/. Source code for benchmarking and applications can be found at https://github.com/cwieder/PathIntegrate_scripts.

## Supporting information

**S1 Supporting Information.** File containing supporting figures and tables**. Fig A in S1 Supporting Information:** Fold changes in COVID-19 multi-omics data based on outcome (mild vs. severe cases)**. Fig B in S1 Supporting Information:** Fold changes in COPDgene multi-omics data based on either COPD status or gender outcomes. **Fig C in S1 Supporting Information: Pathway transformation enhances sensitivity to low signal-to-noise signals (COPDgene semi synthetic data).** Y axis shows proportion of MWU tests significant at Bonferroni p $\leq$ 0.05, performed either on the pathway-level data or the molecular level data, at varying effect sizes shown on X-axis. **Fig D in S1 Supporting Information:** Violin plots showing log10 pathway size for KEGG and Reactome human databases, both for the original databases as well as the database specific coverage (COPDgene and COVID-19). Pathways used are Reactome and KEGG human multi-omics pathways, containing both metabolites and proteins. **Fig E in S1 Supporting Information.** Comparison of PathIntegrate methods classification performance using KEGG and Reactome pathway databases as well as molecular-level model based on semi-synthetic COPDgene data. **Fig F in S1 Supporting Information.** Comparison of PathIntegrate and DIABLO full/sparse models ability to correctly recall target enriched pathway based on semi-synthetic COPDGene data. 'DIABLO pathway (loading)' uses an RGCCA model with no regularisation, whereas 'DIABLO pathway (sparse loading)' uses an RGCCA model with L1 penalty. **Fig G in S1 Supporting Information:** Investigation of effect of sample size in PathIntegrate Single-View (PLS) classification performance on COPDgene data. **Fig H in S1 Supporting Information: Performance of PathIntegrate and DIABLO vs. effect size, based on semi-synthetic data measured by AUROC**. COVID-19 metabolomics and proteomics data were integrated in each model. A. Ability to correctly predict sample outcomes (case vs. control). We compared PathIntegrate Multi-View and Single-View to DIABLO using both molecular and pathway-level multi-omics data. B. Ability to correctly recall target enriched pathway. For 'DIABLO pathway' we compared the full RGCCA model loadings to the sparse model loadings for feature importance. C. Comparison of PathIntegrate Multi-View using KEGG and Reactome pathway databases as well as molecular-level model. D. Effect of sample size on PathIntegrate Multi-View classification performance. For

panels A-C error bars indicate 95% confidence intervals on the mean AUROC (in some cases they appear smaller than point sizes). **Fig I in S1 Supporting Information.** Comparison of PathIntegrate classification performance using KEGG and Reactome pathway databases as well as molecular-level model based on semi-synthetic COVID-19 data. **Fig J in S1 Supporting Information:** Investigation of effects of sample size in PathIntegrate Multi-View (left) and Single-View (PLS) (right) classification performance based on semi-synthetic COVID-19 data. **Fig K in S1 Supporting Information:** 5-times repeated nested 5-fold cross-validated results for number of latent variables parameter tuning in PathIntegrate Multi-View for COPDgene case study integrating metabolomics, proteomics, and transcriptomics data. X axis shows mean AUC across inner folds. Error bars represent standard deviation. **Fig L in S1 Supporting Information:** Preview of PathIntegrate network explorer app (running on a local host server) showing an example of a multi-omics dataset being analysed. Interactive visualisations are facilitated by the open-source Plotly Dash framework (MIT license). Nodes in the network represent pathways and edges represent parent-child relationships between them. Users can zoom in and hover over nodes to see more information about the pathway. **Fig M in S1 Supporting Information:** Reactome hierarchy network (based on coverage in COPDgene multi-omics data) coloured by root pathway membership with full legend. In the interactive app users can hover over nodes to see detailed information about pathway name, root pathway, and coverage in a dataset. **Table A in S1 Supporting Information**: Percentage of molecules with a valid identifier (ChEBI, UniProt, or ENSEMBL) in single omics mapping to Reactome human pathways. A lower percentage of molecules mapping to pathways means a greater percentage of molecules do not yet map to pathways and are not incorporated into pathway-based analyses. **Table B in S1 Supporting Information:** Clinical data definitions for significantly correlated clinical variables from COPDgene study shown in Fig 4F. **Table C in S1 Supporting Information:** Table of notation.
(PDF)

**S1 File. Important multi-omics pathways identified by PathIntegrate Multi-View on COPDgene multi-omics data.** Pathways with p≤0.05 are displayed.
(CSV)

## Acknowledgments

We gratefully acknowledge Peter Castaldi and Craig Hersh for generating the COPDgene RNAseq data.

## Author Contributions

**Conceptualization:** Cecilia Wieder, Rachel PJ Lai, Timothy Ebbels.

**Data curation:** Russell Bowler, Katerina J. Kechris.

**Formal analysis:** Cecilia Wieder.

**Investigation:** Cecilia Wieder.

**Methodology:** Cecilia Wieder, Katerina J. Kechris, Rachel PJ Lai, Timothy Ebbels.

**Project administration:** Timothy Ebbels.

**Resources:** Russell Bowler, Katerina J. Kechris.

**Software:** Cecilia Wieder.

**Supervision:** Rachel PJ Lai, Timothy Ebbels.

**Visualization:** Cecilia Wieder.

**Writing – original draft:** Cecilia Wieder.

**Writing – review & editing:** Cecilia Wieder, Juliette Cooke, Clement Frainay, Nathalie Poupin, Russell Bowler, Fabien Jourdan, Katerina J. Kechris, Rachel PJ Lai, Timothy Ebbels.

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
