## [Decision Letter · Decision Letter 0]

29 Jan 2024

Dear Dr Ebbels,

Thank you very much for submitting your manuscript "PathIntegrate: Multivariate modelling approaches for pathway-based multi-omics data integration" for consideration at PLOS Computational Biology. As with all papers reviewed by the journal, your manuscript was reviewed by members of the editorial board and by several independent reviewers. The reviewers appreciated the attention to an important topic. Based on the reviews, we are likely to accept this manuscript for publication, providing that you modify the manuscript according to the review recommendations.

Sincerely,

Christos A. Ouzounis

Academic Editor

PLOS Computational Biology

Pedro Mendes

Section Editor

PLOS Computational Biology

Reviewer's Responses to Questions

**Comments to the Authors:**

Reviewer #1: This is a valuable piece of work, describing a new approach for multi-omic data integration and providing interpretable and explainable predictions from these datasets through the use of pathways. The work is well described and complete. The experiments are well designed with thorough checks and good comparisons.

There are a couple of limitations to this work which could be further discussed.

1. There is a nice experiment comparing the prediction with the pathintegrate approach, all molecules that are mapped to pathways and finally prediction using all molecules whether mapped or not. I would appreciate additional discussion around the effect of having unmapped molecules, what is the potential impact on the prediction, and also how does this limit the discovery of new biology through this type of anlaysis.

2. When discussing multi-omics and missing data, you might want to look into MOFA which can handle missing data rather well in its models.

Reviewer #2: This manuscript proposes a novel framework (PathIntegrate) for integrating multiple blocks/types of omics data. This framework first reduces the data from the molecular to the pathway level before applying MB-multivariate or classification modelling. The Authors report advantages in signal detection (i.e. elucidating information from low effect sizes) and improved interpretability of results. Overall, the research is technically sound and I recommend that it be published. The comments below are suggested improvements and queries.

1. Figure 2 presents a nice visual summary of the framework after the pathway transformation step is complete. It would be helpful for the readers to see this earlier (at the beginning of the Results) and have the pathway transformation step included. That way, the readers can see the whole framework before they start reading about the details.

2. The link to the tutorials and documentation doesn't work (the link appears to be case sensitive). This direct link may not be needed as it is easy to find the tutorials and documentation through the GitHub page. I commend the authors for including a Google Colab tutorial - it is easy to follow.

3. Lines 105-107, the Authors state that one of the challenges of multi-omics integration is the heterogeneity in the number of features profiled. They go on to say that PathIntegrate addresses this challenge in the pathway transformation step. However, the pathway coverage of the different omics blocks will still exhibit heterogeneity (this is partially observed in Figure 4D). Can the Authors please elaborate on how PathIntegrate resolves this challenge?

4. I observed some Methods content in the Results (example paragraph starting at line 153) and some Discussion content in the Results (examples: lines 173, 282, 340). The Authors may wish to review content placement for improved cohesion.

**Have the authors made all data and (if applicable) computational code underlying the findings in their manuscript fully available?**

Reviewer #1: Yes

Reviewer #2: Yes

PLOS authors have the option to publish the peer review history of their article (what does this mean?). If published, this will include your full peer review and any attached files.

Reviewer #1: **Yes: **Rachel Cavill

Reviewer #2: No

Figure Files:

Data Requirements:

Reproducibility:

References:

---

## [Decision Letter · Decision Letter 1]

11 Mar 2024

Dear Dr Ebbels,

We are pleased to inform you that your manuscript 'PathIntegrate: Multivariate modelling approaches for pathway-based multi-omics data integration' has been provisionally accepted for publication in PLOS Computational Biology.

Best regards,

Christos A. Ouzounis

Academic Editor

PLOS Computational Biology

Pedro Mendes

Section Editor

PLOS Computational Biology

Reviewer's Responses to Questions

**Comments to the Authors:**

Reviewer #2: All queries were addressed.

**Have the authors made all data and (if applicable) computational code underlying the findings in their manuscript fully available?**

Reviewer #2: Yes

PLOS authors have the option to publish the peer review history of their article (what does this mean?). If published, this will include your full peer review and any attached files.

Reviewer #2: No

---

## [Editor Report · Acceptance letter]

21 Mar 2024

PCOMPBIOL-D-24-00040R1 

PathIntegrate: Multivariate modelling approaches for pathway-based multi-omics data integration

Dear Dr Ebbels,

I am pleased to inform you that your manuscript has been formally accepted for publication in PLOS Computational Biology. Your manuscript is now with our production department and you will be notified of the publication date in due course.

With kind regards,

Zsofia Freund
